# An organic artificial cardiomyocyte

Dace Gao [1], Junpeng Ji [1], Simone De Prà [2,3], Miao Xiong [1], Wenlong Jin[1], Ugo Bruno [1], Han-Yan Wu [1], Aleksandr Khudiakov [4], Andreas W. Erhardt [5], Chi-Yuan Yang [1], Peter J. Schwartz [4], Luca Sala [4,6], Iain McCulloch [5,7], Adrica Kyndiah [2], Mario Caironi [2], Magnus Berggren [1,8], Deyu Tu [1] & Simone Fabiano [1,8] ✉

Advances in understanding biological excitability have driven both mathematical modeling and hardware emulation of action potential generation and propagation. While neuromorphic devices based on inorganic or organic systems have advanced rapidly, cardiomorphic hardware remains largely unexplored due to the complexity of reproducing multiple ionic dynamics and the temporal mismatch between the slow cardiac activity and the fast operation of solid-state electronics. Here, we present an organic electrochemical cardiomyocyte (OECM) in which ion-mediated channel currents exhibit time constants aligned with those of ventricular ionic processes. By reproducing a fast sodium current alongside slow, interdependent calcium and potassium currents, the OECM generates ventricular-like action potentials with biorealistic phases, displays refractoriness and responsiveness to electrical or chemical modulation, and synchronizes with bioelectric signals from living cardiomyocytes. These results shift the paradigm of cardiac modeling from purely computational simulations toward biorealistic hardware emulation.

Excitable cells generate action potentials (APs) in response to electrical, chemical, or mechanical stimuli, enabling essential physiological processes including neuronal communication, muscle contraction, and cardiac rhythm regulation[1]. The shape and timing of these APs arise from the interplay of transmembrane ionic currents, and this mechanistic understanding has been central to modern electrophysiology, guiding the development of the Hodgkin-Huxley (HH) model[2] and its derivatives[3,4] as frameworks for neuromorphic simulation[5,6] and hardware implementations[7]. Artificial spiking neurons based on complementary metal–oxide–semiconductor (CMOS) technologies mimic ion-channel–like conductance through subthreshold transistor operation[8,9], but they struggle to reproduce slow ionic dynamics[10] and lack biocompatibility. Organic electrochemical neurons (OECNs)[11–20] overcome many of these constraints. Built from organic mixed ionic-electronic conductors[21], OECNs operate in aqueous environments and exhibit spiking behavior with biologically relevant amplitude, frequency, and energy consumption. Their ionic-gating mechanism enables direct interfacing with living cells[22,23], making them attractive platforms for biointegrated neuromorphic systems.

However, while OECNs provide biologically faithful artificial neurons, an equivalent platform that mimics cardiac excitability is still missing. Ventricular cardiomyocytes (vCMs, Fig. 1a) exhibit multiphase APs shaped by fast Na$^+$ and slow, interdependent Ca$^{2+}$ and K$^+$ currents (Fig. 1b, c). These dynamics are difficult to reproduce with CMOS circuits because biorealistic time constants require large capacitive and resistive components that are incompatible with microelectronics[10]. Organic electrochemical transistors (OECTs), however, intrinsically operate on the same voltage ranges (hundreds of millivolts) and time scales (microseconds to seconds) as vCM ionic

---

[1]Laboratory of Organic Electronics, Department of Science and Technology, Linköping University, Norrköping, Sweden. [2]Center for Nano Science and Technology, Istituto Italiano di Tecnologia, Milano, Italy. [3]Department of Physics, Politecnico di Milano, Milano, Italy. [4]Istituto Auxologico Italiano IRCCS, Center for Cardiac Arrhythmias of Genetic Origin and Laboratory of Cardiovascular Genetics, Milan, Italy. [5]Department of Chemistry, University of Oxford, Oxford, UK. [6]Department of Biotechnology and Biosciences, University of Milano-Bicocca, Milan, Italy. [7]Andlinger Center for Energy and the Environment and Department of Electrical and Computer Engineering, Princeton University, Princeton, NJ, USA. [8]Wallenberg Initiative Materials Science for Sustainability, Department of Science and Technology, Linköping University, Norrköping, Sweden. ✉e-mail: simone.fabiano@liu.se

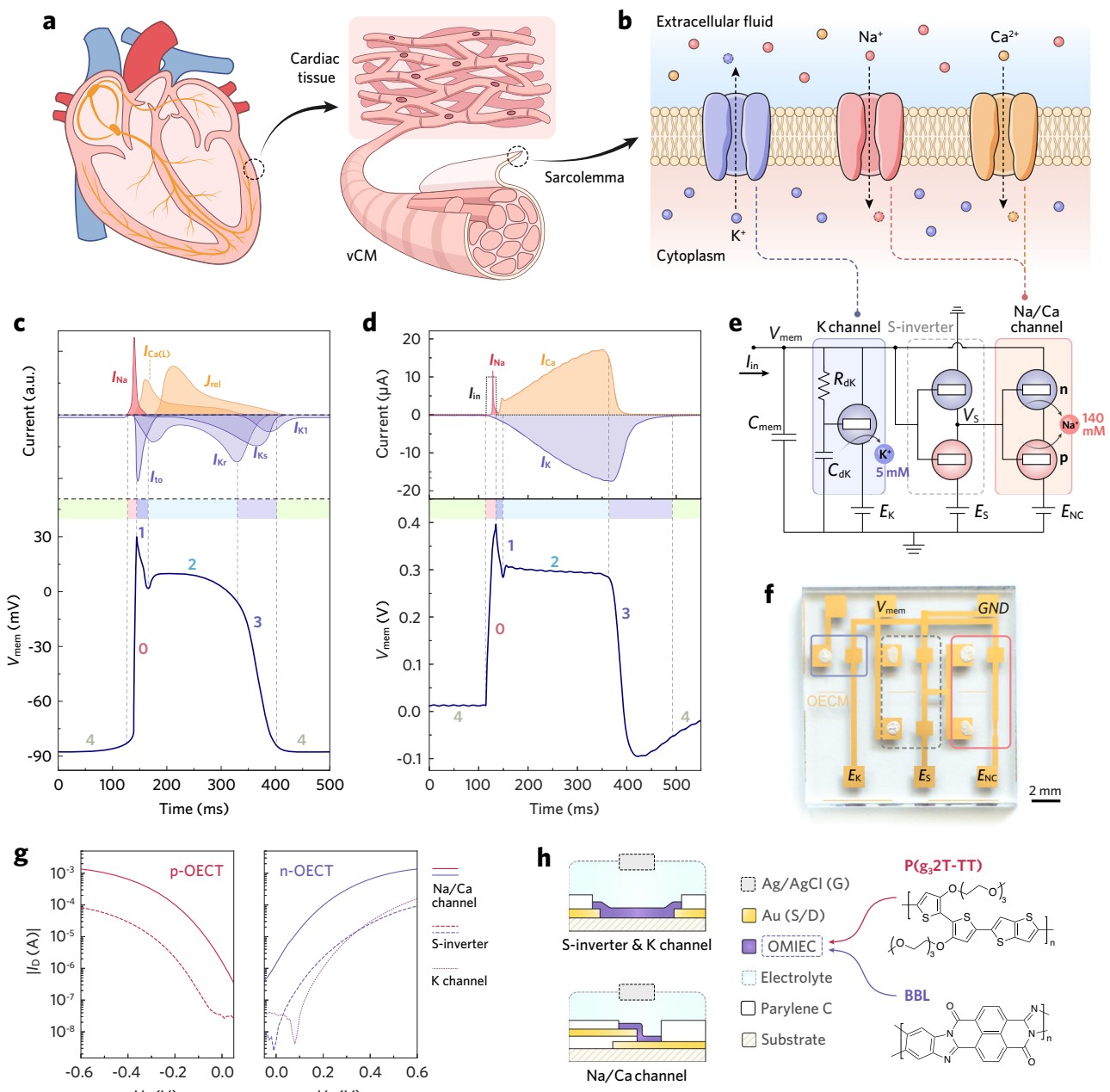

**Fig. 1 | OECM architecture and biophysical plausibility.** Schematic illustrations of **a** a human heart, ventricular cardiac tissue and cardiomyocytes, and **b** ion channels and ionic currents across the sarcolemma of a vCM. **c** Diagram of a typical human ventricular AP profile (bottom) and the time course of multivariate channel currents (top, in arbitrary units), including the fast inward sodium current ($I_{Na}$), L-type calcium current ($I_{Ca(L)}$), calcium release flux ($J_{rel}$), inward rectifier potassium current ($I_{K1}$), transient outward current ($I_{to}$), and delayed rectifier potassium currents ($I_{Kr}$ and $I_{Ks}$). Numbers denote the phases of the vCM AP: rapid depolarization (0), early repolarization (1), plateau (2), repolarization (3), and resting potential (4). Adapted with permission from ref. 55. **d** Diagrams of a typical OECM AP (bottom) and the corresponding channel currents (top). $E_K$ = -0.15 V. Numbers denote the phases of the OECM AP: rapid depolarization (0), early repolarization (1), plateau (2), repolarization (3), and resting potential (4). **e**, Equivalent circuit of the OECM. Standard testing condition: $E_{NC} = E_S = 0.6$ V, $C_{mem} = 1\,\mu F$, $C_{dK} = 0.1\,\mu F$, $R_{dK} = 470\,k\Omega$, [NaCl] = 140 mM, [KCl] = 5 mM, $I_{in} = 10\,\mu A$, 20 ms. $E_K$ varies between −0.05 and −0.15 V to accommodate device-to-device variations, enabling a standard APD of ~300 ms across different OECM chips. **f** Photograph of an OECM chip. **g** Transfer characteristics of the constituent OECTs in a typical OECM. **h** Device architectures (left) and molecular structures of the channel materials (right) used in the constituent OECTs.

processes due to their volumetric capacitance and ion-doping kinetics[24,25]. This makes OECTs uniquely suited for cardiomorphic hardware and positions them as a promising platform for creating artificial cardiomyocytes capable of real-time operation and direct interfacing with biological tissue.

Here, we introduce the organic electrochemical cardiomyocyte (OECM), a cardiomorphic wetware that replicates ventricular AP, including upstroke, notch, plateau, and repolarization phases, with biorealistic temporal and amplitude characteristics (Fig. 1d). By mapping OECT conductance changes onto the major vCM channel currents, the OECM emulates three distinct ionic processes and responds predictably to electrical or chemical modulation. Moreover, OECMs can be paced by bioelectrical activity from living cardiomyocytes, demonstrating communication and enabling hybrid synchronization

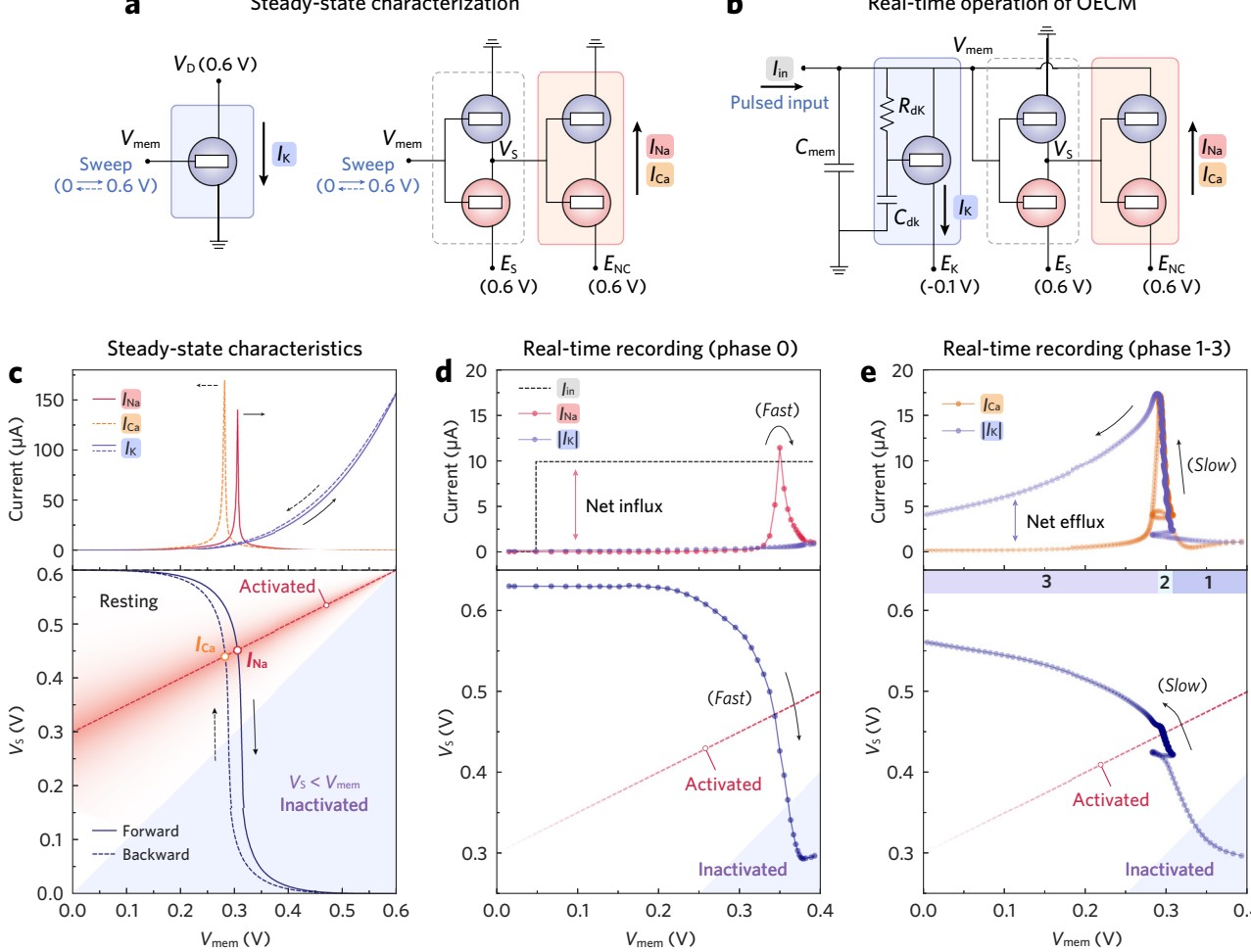

**Fig. 2 | Steady-state and real-time characterization of the OECM. a** Schematic circuits illustrating the testing conditions for steady-state characterization. **b** Schematic circuit illustrating the testing condition for real-time OECM operation. **c** Steady-state characteristics of the K channel and Na/Ca channel. The K channel OECT was evaluated by its transfer curve ($I_K$, top), measured at $V_D = 0.6$ V while sweeping $V_{mem}$ from 0 to 0.6 V and back to 0 V. The measurement was performed without $R_{dK}$ and $C_{dK}$ to reveal the intrinsic device behavior. The cascaded S-inverter was evaluated by its voltage transfer curve ($V_S$, bottom), while the Na/Ca channel was evaluated by its current-transfer curve ($I_{Na}$, $I_{Ca}$, top). $E_S$ and $E_{NC}$ were fixed at 0.6 V, while $V_{mem}$ was swept from 0 to 0.6 V and back to 0 V. The red dashed line

denotes $V_S = (V_{mem} + E_{NC})/2$, corresponding to full activation of the Na/Ca channel. **d**, **e** Transient response of the operating OECM. Channel currents (top) and S-inverter output (bottom) are plotted as a function of $V_{mem}$. $E_S$, $E_{NC}$, and $E_K$ were fixed at 0.6 V, 0.6 V, and −0.1 V, respectively. An $I_{in}$ pulse triggered the OECM, and the evolution of $V_{mem}$, $V_S$, $I_{Na}$, $I_{Ca}$, and $I_K$ was recorded. Data were sampled at 0.5 ms intervals, so the point density reflects the rate of change of $V_{mem}$. In phase 0 (**d**), the low point density near $V_{mem} \approx 0.3$ V indicates a rapid sweep and transient $I_{Na}$ activation. In phase 2 (**e**), the higher point density near $V_{mem} \approx 0.3$ V reflects prolonged activation of both Na/Ca and K channels.

between artificial and biological excitability. Notably, the OECM models the electrophysiological function of a vCM rather than its mechanical actuation, distinguishing it from artificial-muscle technologies[26,27].

## Results

### OECM architecture and analogy to vCMs

The OECM is designed following a framework analogous to the Beeler-Reuter (BR) model[28] (Supplementary Note 1), the first comprehensive mathematical description of vCM excitability. The BR model incorporates a fast sodium current as well as slow calcium and potassium dynamics, capturing the prolonged AP plateau phase essential for excitation-contraction coupling[29]. While subsequent models, most notably the Luo–Rudy dynamic model[30,31], refined ionic kinetics and calcium handling for greater biological realism, the BR model remains the minimal, biophysically grounded reference that bridges experimental[32] and computational[33,34] cardiophysiology.

The OECM creates a physical implementation of the BR model (Supplementary Note 2) by associating the membrane capacitance ($C_{mem}$) with the sensing, charging, and discharging blocks at the membrane-voltage node ($V_{mem}$; Fig. 1e, f). The behavior of the charging and discharging blocks is governed by the device physics of the OECTs. When all OECTs are off, the channel currents are minimal, and $V_{mem}$ remains at the resting potential. When the OECTs in the charging Na/Ca channel switch on, the high potential bias ($E_{NC}$, 0.6 V) supplies current to $C_{mem}$, increasing $V_{mem}$. Conversely, when the OECT in the discharging K channel turns on, the low potential bias ($E_K$, around −0.1 V) discharges $C_{mem}$, reducing $V_{mem}$. The Na/Ca channel is gated by the output of the sensing inverter (S-inverter), which acts as a voltage-triggered switch that continuously monitors $V_{mem}$ and activates the Na/Ca channel when a threshold is reached. In contrast, the K channel is gated directly by $V_{mem}$, with an RC element coupled to its gate to slow the response. The resulting channel currents integrate across $C_{mem}$, thereby modulating $V_{mem}$. The temporal evolution and relative magnitude of the charging and discharging currents (Fig. 1d, top)

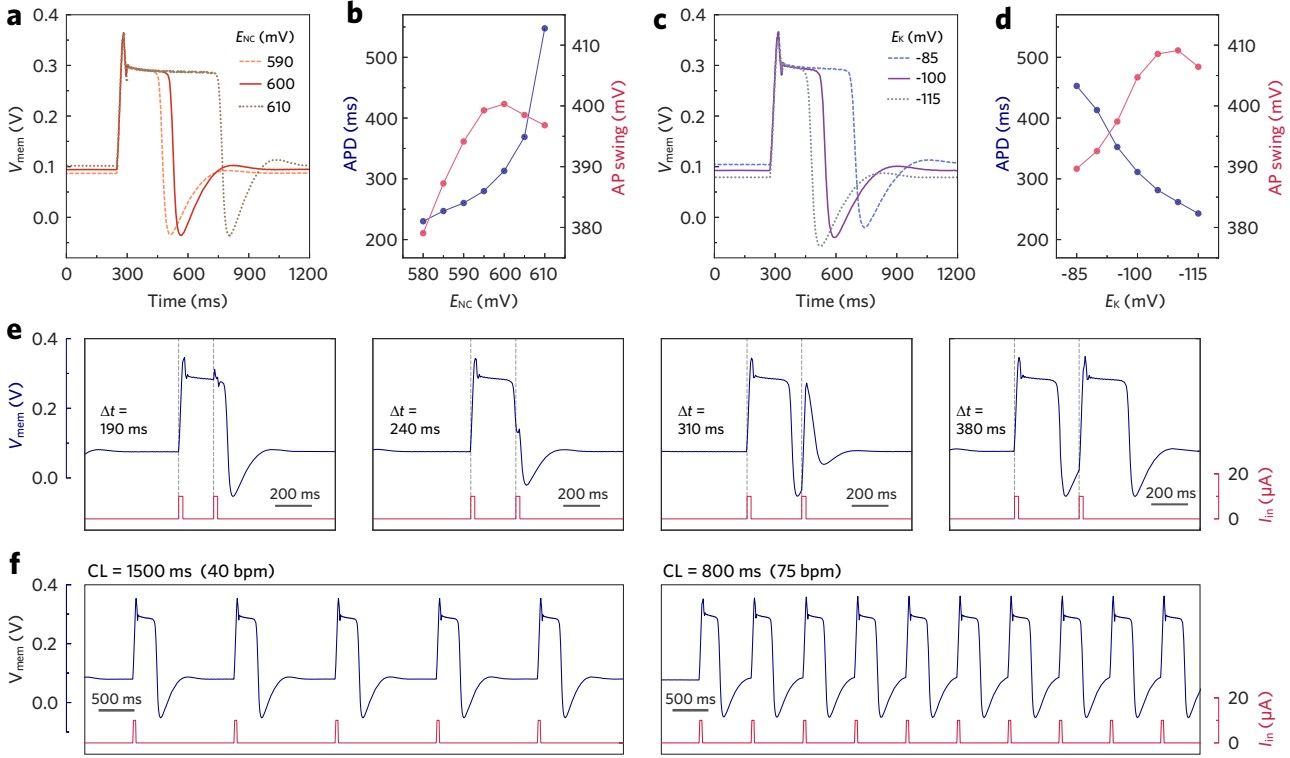

**Fig. 3 | Electrical modulation of OECM. a, b** Comparison of OECM AP profiles in APD (blue dots) and swing amplitude (red dots) under different $E_{NC}$ biasing conditions; $E_K = -0.11$ V. **c, d** Comparison of OECM AP profiles in APD (blue dots) and swing amplitude (red dots) under different $E_K$ biasing conditions. **e** Characterization of OECM refractoriness with varying intervals between two consecutive $I_{in}$ pulses; $E_K = -0.12$ V. **f** Recordings of OECM AP trains with different cycle lengths; $E_K = -0.12$ V, CL: cycle length.

produce ventricular-shaped OECM APs (Fig. 1d, bottom). Below, we describe these circuit blocks in detail and discuss their steady-state characteristics (Fig. 2a) as well as transient responses (Fig. 2b).

The charging Na/Ca channel is implemented using an OECT-based inverter (Supplementary Fig. 1). BBL and P(g₃2T-TT) serve as channel materials for the n-type and p-type accumulation-mode OECTs, respectively, owing to their matched threshold voltage ($V_{th}$) and drive strengths[20,35] (Fig. 1g, h). Controlled by its input ($V_S$) and supply (high: $E_{NC} = 0.6$ V, low: $V_{mem}$) voltages, this dual-OECT configuration produces an antiambipolar charging current[13] that peaks at the transition voltage when both OECTs operate in saturation. Stair-type, high-current OECTs are employed in the Na/Ca channel to ensure sufficient charging capacity during OECM operation. The application of a biorealistic NaCl concentration (140 mM, physiological level of extracellular sodium ions ([Na⁺]ₒ)) in the OECTs' gating electrolyte allows the Na/Ca channel to switch promptly ($\tau_{on} \approx 0.17$ ms, Supplementary Fig. 3a) in response to $V_S$ modulation.

Preceding the Na/Ca channel is the S-inverter (Supplementary Fig. 2), which inverts $V_{mem}$ into $V_S$ to gate the Na/Ca channel. It responds rapidly to changes in $V_{mem}$ ($\tau_{on} \approx 0.21$ ms, Supplementary Fig. 3b), serving as a threshold detector that regulates the Na/Ca channel's conductance. Cascading the S-inverter before the Na/Ca channel reshapes the antiambipolar current response, yielding a sharp peak in the Na/Ca channel's current-transfer characteristics (CTC, Fig. 2c, top). With fixed voltage supplies (high: $E_S = 0.6$ V, low: GND), the S-inverter switches at $V_{mem} \approx 0.3$ V, turning on the charging channel as $V_S$ sweeps through the midpoint of its effective power rail ($V_S = (V_{mem} + E_{NC})/2 \approx 0.45$ V). The relative potential of $V_S$ with respect to $V_{mem}$ determines the Na/Ca channel's tristate behavior (Fig. 2c, bottom, see Supplementary Figs. 4 and 5 for details). Under a steady-state $V_{mem}$ sweep, the Na/Ca channel subsequently transitions through

resting, activated, and inactivated states during the forward sweep, and reverses these transitions over the backward sweep. The corresponding charging currents are denoted as $I_{Na}$ and $I_{Ca}$, respectively. Although similar in steady-state CTCs, their transient characteristics during OECM operation differ significantly depending on the discharging conditions.

The discharging K channel comprises a planar BBL-OECT (Fig. 1g, h), with its source negatively biased by $E_K$ at around $-0.1$ V. The application of a low-concentration electrolyte (5 mM KCl, physiological level of extracellular potassium ions ([K⁺]ₒ)) elevates its threshold voltage ($V_{th}$) to 0.29 V (Supplementary Fig. 6), while concurrently impairing its switching speed ($\tau_{on} \approx 16.2$ ms, Supplementary Fig. 7). Additionally, the RC element ($R_{dK}$-$C_{dK}$) coupled at its gate further introduces a circuitry-level delay. By properly assigning the passive components (resistors, capacitors) and optimizing the biasing conditions (facilitated by SPICE simulation, Supplementary Figs. 8–11), the OECM is regulated so that the K channel remains closed during the forward swing and gradually opens during the backward swing. Instead of constructing separate Na and Ca channels, the OECM leverages the forward and backward dual swings of a single charging channel, together with the voltage-time-dependent discharging through the K channel, to deliver both fast $I_{Na}$ and sustained $I_{Ca}$ at designated AP phases. This approach enables the OECM to generate cardiomorphic APs without complicating the circuitry. A comprehensive comparison between the OECM and existing OECN architectures is provided in Supplementary Note 3.

## AP phase analysis

The real-time operation of an OECM depends not only on the steady-state behavior of its building blocks but also on their transient characteristics across different AP phases. In living cardiac tissues, a resting

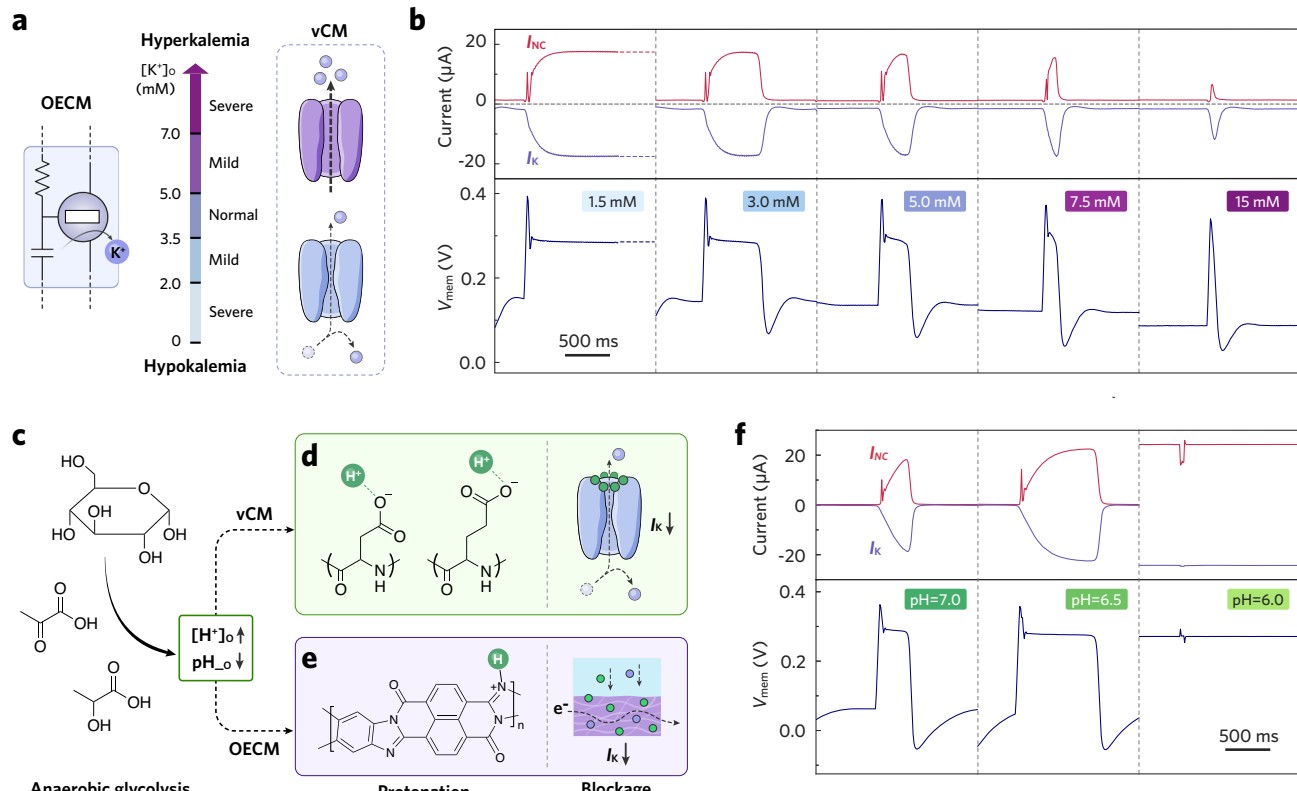

**Fig. 4 | Chemical modulation of OECM. a** Schematic illustrations detailing the characteristics of hyperkalemia and hypokalemia, highlighting the normal and abnormal ranges of K⁺ concentration and the corresponding behavior of K⁺ channels in vCMs. **b** OECM APs (bottom) and channel currents (top) recorded with different [KCl] in the K channel OECT's gating electrolyte; $E_K = -0.09$ V. Schematic illustrations depicting the features of lactic acidosis, including **c** the reaction pathway for lactic production ([H⁺]$_o$: extracellular proton concentration, pH$_{\_o}$: extracellular pH), and the influence of increased proton concentrations on **d** K⁺ channels in vCMs and on **e** the K channel in OECMs. Note that panel (e) illustrates one representative proton-BBL association motif and does not imply a unique binding configuration. **f** OECM APs (bottom) and channel currents (top) recorded at different pH levels; $E_K = -0.14$ V.

vCM gets activated when receiving a rapid electrotonic current through gap junctions[36] from an adjacent, depolarized vCM. Similarly, an OECM is stimulated upon receiving an input current pulse ($I_{in}$) at its $V_{mem}$ node. During phase 0, the $I_{in}$ pulse charges the membrane capacitance ($C_{mem}$), resulting in a rapid ascent of $V_{mem}$ from its resting potential to -0.3 V. Consequently, the S-inverter switches and sweeps $V_S$ across -0.45 V quickly (Fig. 2d, bottom), which transiently activates the charging Na/Ca channel (-15 ms) and yields a rapid $I_{Na}$ spike that accelerates depolarization. Throughout the forward $V_{mem}$ swing, the outward $I_K$ remains low (<1 μA) due to the delayed activation of the K channel. The net inward charge from $I_{Na}$ (Fig. 2d, top) further propels $V_{mem}$ up to -0.39 V, yielding the archetypal upstroke.

Phase 1 is marked by the notch, i.e., early repolarization, immediately following the upstroke. With $V_{mem}$ staying above 0.3 V, the Na/Ca channel becomes temporarily inactivated ($V_S < V_{mem}$) while the K channel gradually opens—akin to the behavior of slow delayed rectifier K⁺ channels[30] in vCMs—and brings $V_{mem}$ down to -0.3 V. At the onset of phase 2, both channels partially open and establish a balanced charging-discharging strength at -5 μA. This delicate balance between $I_{Ca}$ and $I_K$ (Fig. 2e, top) results in a secondary equilibrium (Supplementary Note 4) of OECM at $V_{mem} \approx 0.3$ V ($V_S \approx 0.45$ V, Fig. 2e, bottom), manifested as a plateau that persists for -200 ms. Herein, the activation of the discharging K channel extends the activation of $I_{Ca}$. Throughout the plateau, $I_K$ and $I_{Ca}$ increase gradually until 17.2 μA in a synchronized manner. By the end of phase 2, the strength of $I_K$ surpasses $I_{Ca}$, bringing $V_{mem}$ down below 0.3 V. This, in turn, deactivates the Na/Ca channel promptly, while the K channel deactivates at a slower rate. The resultant net discharge (Fig. 2e, top) triggers

repolarization and hyperpolarization (phase 3), further shutting down both channels. Ultimately, the OECM reverts to its primary equilibrium at the resting potential (phase 4), where the currents find balance once again at a low strength of 0.2 μA. The above analysis implies that the conductances of the OECM's Na/Ca and K channels are interdependent. This closely mirrors the interdependence of K⁺, Na⁺, and Ca²⁺ currents across the sarcolemma of vCMs[30].

## Electrical modulation

The OECM exhibits biorealistic AP features that resemble those of vCMs. Notably, the generation of a complete OECM AP necessitates an adequate $I_{in}$ stimulus. Insufficient charge delivery results in a sub-threshold spike, whereas excessive charging prevents repolarization (Supplementary Fig. 12). A square pulse of $I_{in}$ set at 10 μA for 20 ms provides a suitable charging rate and sufficient charge supply, thus is employed as the standard stimulus for OECM operation. The charging rate also influences the upstroke velocity (d$V$/d$t$) during phase 0, where a higher $I_{in}$ amplitude leads to a steeper depolarization trajectory (Supplementary Fig. 13). Additionally, the duration of the phase 2 plateau—a pivotal factor in vCMs for excitation-contraction coupling—can be finely tuned between 200 and 500 ms with subtle adjustments on the biasing voltages. A more positive $E_{NC}$ enhances the Na/Ca channel's conductance upon activation, which helps to maintain the secondary equilibrium at $V_{mem} \approx 0.3$ V and thereby prolongs the plateau (Fig. 3a, b). In contrast, a more negative $E_K$ enhances the discharging strength of $I_K$, making the equilibrium less stable and consequently shortening the plateau (Fig. 3c, d). The electrical tunability of action potential duration (APD) enables the OECM to emulate

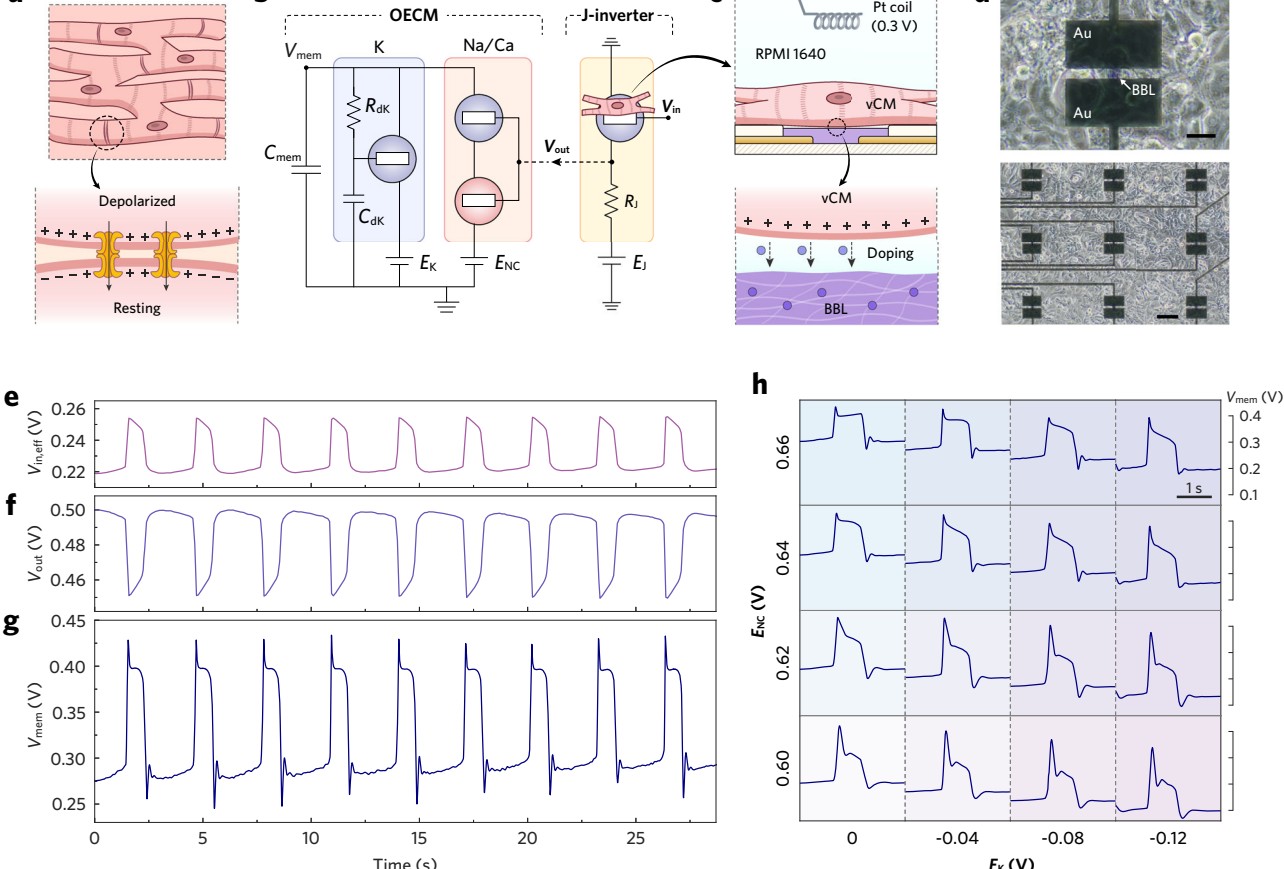

**Fig. 5 | Synchronization of OECM AP with biotic cardiac APs. a** Schematic illustrations of the electrical coupling between vCMs via gap junctions. **b** Equivalent circuits of OECM and J-inverter, highlighting the bioelectrical signaling pathway from hiPSC-CMs, through the J-inverter, to the OECM. $V_{in} = 0.3$ V, $E_J = 0.6$ V, $R_J = 180$ kΩ. **c** Schematic illustrations depicting the interface between a hiPSC-CM and a BBL channel. **d** Microscopic photographs showing the configuration of the biohybrid OECT, where a monolayer of hiPSC-CMs was cultured on top of a BBL channel (top; scale bar, 30 μm). The biohybrid OECT chip consists of an array of BBL channels

(bottom; scale bar, 100 μm), and the channel with optimal hiPSC-CM-to-OECT coupling was utilized to build the J-inverter. **e, g** Side-by-side plots of voltage trains at critical circuitry nodes upon the synchronization of APs from hiPSC-CMs to the OECM. **f** $V_{out}$ was recorded in vitro as the J-inverter's output in response to APs generated by hiPSC-CMs. **g** $V_{mem}$, i.e. the OECM's AP train, was recorded in response to $V_{out}$ stimulation. **e** $V_{in,eff}$ was back-calculated from $V_{out}$. **h** A matrix of OECM APs displaying variant profiles produced under different biasing conditions.

vCMs from different ventricular regions (epi-, mid, and endocardial cells), wherein vCMs with lower K$^+$ channel density exhibit reduced $I_K$ and prolonged APD[37].

The OECM also reproduces refractoriness (Fig. 3e, Supplementary Fig. 14), a vital mechanism for vCMs to prevent tetanic contraction. Under the standard operating condition (discussed in "Methods" section), the OECM has an absolute refractory period (ARP) of ~240 ms, spanning from phase 0 to early phase 3, during which no AP can be initiated. ARP is succeeded by a relative refractory period (RRP, lasting ~80 ms), during which an $I_{in}$ pulse can only elicit a premature spike. The characteristics of such spikes are contingent upon the timing of the stimulus: the later the stimulation occurs within RRP, the greater the amplitude that the spike can attain. The OECM regains full excitability after 320 ms, permitting the generation of a subsequent full AP.

Under periodic $I_{in}$ stimulation, the OECM produces rhythmic APs akin to heartbeats, with the cycle length defined by the interval between consecutive input pulses (Fig. 3f). Continuous pacing at 60 beats per minute for 1 h maintained stable AP generation, with the AP waveform largely preserved despite moderate changes in amplitude and duration (~20%, Supplementary Fig. 15). These changes in the AP profile primarily originate from gradual degradation of P(g$_3$2T-TT), likely due to oxygen-induced side reactions[20]. When direct current $I_{in}$ is

supplied, the OECM produces non-resting and rapidly pacing APs without a diastolic interval, where the AP features are susceptible to $I_{in}$ amplitude and biasing conditions (Supplementary Figs. 16 and 17).

## Ionic and pH modulation

In cardiac tissues, the maintenance of a normal cardiac rhythm demands a closely regulated physiological environment, encompassing factors like temperature, electrolyte concentration, pH, etc. For example, [K$^+$]$_o$ in human bodies should be tightly regulated within the range from 3.5 to 5.0 mM. A deviation from this range can disrupt homeostasis and create the substrates that are prone to arrhythmias (Fig. 4a). Hyperkalemia[38] ([K$^+$]$_o$ > 5 mmol/L) shortens a vCM's APD due to an allosteric effect that exaggerates the K$^+$ efflux. Conversely, hypokalemia[38] ([K$^+$]$_o$ < 3.5 mmol/L) induces the opposite effects against hyperkalemia.

Our previous study reveals that the transconductance of a BBL-channeled OECT is positively correlated with the ionic strength of its gating electrolyte[11] (Supplementary Fig. 18b). This enables the OECM to replicate hyper- and hypokalemia by adjusting the electrolyte concentration applied to the K channel OECT (Fig. 4b, Supplementary Fig. 19). A reduction in [KCl] suppresses discharging current, thereby reducing the repolarization reserve. Consequently, this prolongs the

OECM's APD at [KCl] = 3 mM and induces sustained depolarization at [KCl] = 1.5 mM. In contrast, an increase in [KCl] can effectively shorten the APD at [KCl] = 7.5 mM, and give rise to premature excitation—manifesting as a spike without plateau—at [KCl] = 15 mM. Likewise, the Na/Ca channel is also susceptible to ionic modulation, emulating the phenomenon of hyper- and hyponatremia[39] (Supplementary Figs. 20 and 21). The influence of ionic strength on OECM AP profiles is summarized quantitatively in Supplementary Tables 1 and 2.

Beyond electrolytic imbalances, deviations in extracellular pH from the physiological range (7.35–7.45)—even slight ones—can induce life-threatening dysfunctions in the cardiac system. For instance, during cardiac hypoxia associated with acute ischemia, the enhanced anaerobic glycolysis converts glucose through pyruvate to lactic acid, leading to metabolic acidosis (Fig. 4c). The excessive protons can protonate amino acid residues within various $K^+$ channels and thus obstruct $K^+$ passage (Fig. 4d), which will eventually extend vCMs' APD and increase the risk of arrhythmias[40]. The OECM exhibits a comparable pH sensitivity because BBL is prone to protonation-induced current inhibition[41] (Fig. 4e, Supplementary Fig. 18c). With a constant [KCl] in the gating electrolyte of the K channel, modulating pH from neutral (7.0) to mildly acidic levels can markedly diminish channel conductance, leading to a prolonged APD at pH = 6.5, and even failure of depolarization at pH = 6.0 (Fig. 4f, Supplementary Fig. 22).

### Synchronization of biotic APs

The potential of a cardiomorphic device can be further exploited if it can couple to living vCMs and synchronize its artificial APs with bioelectrical activities. Herein, we developed a strategy to facilitate vCM-to-OECM synchronization by referring to the gap-junction-mediated conduction observed in living vCMs. In this natural context, an activated vCM depolarizes adjacent resting cells by electrotonic interaction, propagating the AP across the tissue (Fig. 5a). To mimic this process, a junctional inverter (J-inverter, Supplementary Fig. 23) was established between a monolayer of human induced pluripotent stem cell-derived cardiomyocytes (hiPSC-CMs) and an OECM (Fig. 5b), where the hiPSC-CMs directly interface with the BBL channel in the electrolytic environment of an OECT (Fig. 5c, d). The long-term reliability of this configuration is attributed to the biocompatibility, cell viability, and chemical stability of BBL[42], which allows the RPMI 1640 medium to serve as both the cell culture environment and the electrolyte for the BBL-channeled OECT (Supplementary Fig. 24a). The biohybrid OECT functions under the gating control of a platinum coil, with an input voltage ($V_{in}$) set at 0.3 V to establish an appropriate transconductance. The biotic APs generated by the hiPSC-CMs act as a secondary alternating voltage bias that superimposes onto $V_{in}$ and further modulates the OECT upon the firing of APs[23,43]. Note that only a fraction of the gating potential can effectively couple to the BBL channel due to the damping impedance existing in the ionic circuit of the OECT (Supplementary Fig. 24b). The J-inverter was constructed by connecting the biohybrid OECT in series with a load resistor ($R_l$), through which the upward hiPSC-CM APs were amplified and translated into downward $V_{out}$ pulses (Fig. 5f) suitable for OECM modulation. Compared to the S-inverter, the lower voltage gain of the J-inverter results in a smaller $V_{out}$ swing and therefore weaker modulation of the Na/Ca channel conductance. Controlled experiments (Supplementary Fig. 25) confirm that the $V_{out}$ amplitude strongly influences the OECM AP shape, suggesting that the J-inverter could be further optimized by increasing the transconductance of the biohybrid OECT. The effective input voltage ($V_{in,eff}$, Fig. 5e) coupling to the biohybrid OECT can be back-calculated through the standard voltage transfer characteristics (VTC) of the J-inverter.

The OECM driven by bioelectric signals is simplified to a combination of Na/Ca and K channels, as the tristate behavior of the Na/Ca channel is directly governed by the output of the J-inverter. By feeding the in vitro recording of $V_{out}$ into the OECM circuit, we obtained a series of stochastic OECM APs (Fig. 5g) that exhibit a close synchronization with the $V_{out}$ and $V_{in,eff}$ pulses. This stochasticity contrasts with the deterministic APs generated by self-regulated OECMs (Fig. 3f) and is interpreted as an amplification of the inherent stochasticity from hiPSC-CM APs. Such findings suggest that the OECM has the potential to communicate with diseased vCMs, such as those affected by ventricular fibrillation, while reproducing their stochastic dynamics. Furthermore, we demonstrated the flexibility in modulating the profile of OECM APs by adjusting the biasing conditions, where variations in the ratio of charging-discharging strength could yield a matrix of AP waveforms (Fig. 5h) with alternations in resting potential, upstroke amplitude, notch size, plateau slope, and extent of hyperpolarization.

## Discussion

Leveraging the ion-dependent switching characteristics of OECTs, we developed biorealistic OECMs that reproduce the interdependent Na/Ca and K channel dynamics and generate cardiomorphic APs with phasic profiles closely resembling those of native vCMs. This advance fills a longstanding gap in cardiomorphic hardware by extending cardiac electrophysiology modeling beyond purely computational domains into physical, hardware-based emulation. OECMs exhibit intrinsic sensitivity to ionic concentrations in the electrolyte, enabling the modeling of pathological states such as long QT syndrome[44] or hyper/hypokalemia. Furthermore, the observed synchronization of APs between hiPSC-CMs and OECMs marks an essential first step toward direct bioelectronic entrainment between artificial and biological cardiac systems. Future directions will focus on establishing bidirectional interfaces between OECMs and living cardiomyocytes, enabling OECM-to-vCM stimulation and forming closed-loop biohybrid signaling systems in vitro and potentially in vivo. Such systems could enable OECM-based pacing to modulate the rhythmic activity of contractile organoids[45,46], guide the maturation or retraining of engineered cardiac tissue[47], or restore electrical conduction across infarcted or fibrotic regions. Beyond pacing, OECMs could serve as autonomous sensors for detecting early ectopic or reentrant electrical activity, delivering adaptive counter-stimulation to restore synchrony within diseased myocardium.

Translation from a rigid OECM chip toward implantable systems will require addressing several challenges. Long-term biocompatibility and electrochemical stability must be ensured in the dynamic, ion-rich cardiac environment, where chronic inflammation, fibrotic encapsulation, and material degradation could compromise both function and safety. This requires the OECMs to adopt solid-state electrolytes, robust encapsulation strategies, and more chemically stable organic semiconductors to replace degradation-prone materials. Mechanically, OECMs must transition from glass substrates to low-modulus materials that can accommodate the myocardium's continuous cyclic deformation while maintaining stable electrical coupling. In addition, chronic operation will require ultra-low-energy architectures compatible with implantable power constraints. Addressing these challenges will be essential for advancing OECM platforms from in vitro biohybrid systems toward safe and effective in vivo deployment.

While the present OECM architecture emulates a single vCM, we envision that further miniaturization and integration could yield OECM-based analog emulators capable of reproducing cardiac conduction at the tissue level. Conventional digital simulations often rely on homogenized models[48] that approximate cardiac tissue as a continuous medium with averaged properties. Although computationally efficient, these models inherently obscure single-cell behavior and fail to capture the impact of local heterogeneity[49]. In contrast, the envisaged large-scale OECM arrays would preserve discrete and tunable cellular identities, while allowing for controlled implementation of graded electrical or ionic heterogeneity across individual units. This unique capability could facilitate real-time, parallel investigations of tissue-level pathological processes—such as unidirectional block or

reentrant circuits—that remain difficult to access with continuum models or to reproduce consistently in biological experiments.

## Methods

### Materials
BBL[50] and P(g$_3$2T-TT)[51] were synthesized by following previously reported routes. Sodium chloride (NaCl), potassium chloride (KCl), lactic acid (HLac), sodium bicarbonate (NaHCO$_3$), 1,1,2,2-tetra-chloroethane, methanesulfonic acid (MSA), and 3-(Trimethoxysilyl) propyl methacrylate (Silane A174) were purchased from Sigma-Aldrich and used as received.

### OECM fabrication
OECM chips were fabricated via photolithography. Glass wafers were cleaned by sonication in soap water (2% Micro-90 in deionized water), acetone, and isopropyl alcohol, then dried with nitrogen. To prepare OECT electrodes, interconnects, and contact pads (**M1**), 5 nm chromium and 50 nm gold were thermally evaporated (T090M, Moorfield) onto the substrates and patterned by lift-off. A 0.8-µm-thick Parylene C layer (**P1**) was then deposited by chemical vapor deposition (CVD, Diener), serving as an insulating layer that prevents parasitic capacitance between metal and electrolytes, as well as a vertical separator whose thickness defines the channel length of the stair-type OECTs in the Na/Ca channel. **P1** was selectively removed by reactive ion etching (RIE, Etchlab 200, Sentech), where the etching area was defined by a layer of patterned photoresist. The second metal layer (**M2**) was deposited via thermal evaporation and patterned by wet etching, which contacts **M1** through the opening of **P1** and serves as the top source electrodes for the stair-type OECTs. A double-layer Parylene C (**P2**, 0.45/0.8 µm) was deposited in sequence by CVD, with an ultrathin layer of Micro-90 (2% in deionized water) spin-coated in between two layers as an anti-adhesive agent. The bottom layer of **P2** protects **M2**, while the top layer serves as the sacrificial layer for OMIECs patterning. **P2** was selectively removed by RIE to define the channel areas. A 6-µm-thick positive photoresist (AZ-10XT 520 cp) was used as protection for **P1** and **P2** dry etching, while a 1.3-µm-thick positive photoresist (S1813) was employed for wet etching and lift-off. All photoresists were patterned using a mask aligner (MA6, SUSS) through a series of photomasks. Finally, the OECM chips were diced from the processed glass wafer using a laser cutter (Metaquip). N-type OECTs on OECM were prepared by spin coating a BBL solution (2.5 mg/mL in MSA) within a designated chip area, followed by soaking in water for MSA removal, and drying with nitrogen for film formation. P-type OECTs were prepared by spin coating a P(g$_3$2T-TT) solution (3 mg/mL in 1,1,2,2-tetra-chloroethane) within a designated chip area. The OMIECs were patterned within the OECT channels by peeling the sacrificial layer of **P2**. Ag/AgCl paste was applied manually to all the OECT gate electrodes and dried with nitrogen to complete an OECM chip.

### Channel dimensions
The OECTs within OECM are designed with different channel dimensions to ensure a matched drive strength between the p-n OECT pairs in the S-inverter and Na/Ca channel, and to achieve proper charging-discharging currents in K and Na/Ca channels. The geometry of all OECTs in OECM is listed as follows: (1) K channel, n-type: $W = 600$ µm, $L = 6$ µm. (2) S-inverter, n-type: $W = 200$ µm, $L = 6$ µm. (3) S-inverter, p-type: $W = 100$ µm, $L = 12$ µm. (4) Na/Ca channel, n-type: $W = 800$ µm, $L \approx 0.8$ µm. (5) Na/Ca channel, p-type: $W = 400$ µm, $L \approx 0.8$ µm. BBL in all n-type OECTs has a thickness of ~20 nm, while P(g$_3$2T-TT) in all p-type OECTs has a thickness of ~5 nm.

### Electrical characterization
A parameter analyzer (Keithley 4200A-SCS, Tektronix) was employed to characterize the OECTs and inverters in OECMs, and to perform regular recording of OECM APs. The switching speed of the K and Na/

Ca channels was measured using an ultra-fast I-V module (4225-PMU, Tektronix) integrated within the parameter analyzer. The high-resolution sampling of an OECM AP (Figs. 1h and 2d, e) was performed by using a high-precision, high-density source measure unit (PXIe-4163, National Instruments) with a sampling interval of 0.5 ms. The standard testing condition for the OECM is defined to facilitate the generation of APs with a duration of ~300 ms. Specifically, $E_{NC}$ and $E_S$ are held constant at 0.6 V. $C_{mem}$, $C_{dK}$, and $R_{dK}$ are 1 µF, 0.1 µF, and 470 kΩ, respectively. The $I_{in}$ pulse is set at 10 µA for a duration of 20 ms. The electrolyte utilized for the Na/Ca channel and the S-inverter comprises 140 mM NaCl dissolved in deionized water, while the electrolyte for the K channel comprises 5 mM KCl dissolved in deionized water. $E_K$ is strategically adjusted within the range of −0.05 to −0.15 V. This calibration accommodates device-to-device variations, ensuring that each OECM can be benchmarked with an APD of approximately 300 ms prior to further testing. All OECM AP recordings in this work were conducted under the standard testing conditions, unless otherwise specified.

### SPICE simulation
The SPICE models of OECM and all its constituent OECTs were created in QSPICE Simulator (Qorvo). The OECT models were constructed with reference to a previously reported circuitry design[11] (Supplementary Fig. 8), with electrical parameters modified to match the steady-state and transient OECT characteristics in experimental results. The OECM model was constructed by integrating the OECT models alongside passive components according to the real-world OECM design (Supplementary Fig. 10). OECM operation and electrical modulation were simulated directly in QSPICE, with great flexibility in the adjustment of the biasing and stimulating conditions.

### Cell culture and differentiation
Bona fide healthy human induced pluripotent stem cells (hiPSCs) were obtained from the Coriell Institute for Medical Research (identifier GM25256; WTC-11 line, hPSCReg UCSFi001-A) under an appropriate Material Transfer Agreement. HiPSCs were maintained on 6-well multiwell plates coated with recombinant human vitronectin (rhVTN, Thermo Fisher Scientific) in Essential 8 (E8) Flex medium (Thermo Fisher Scientific). For cardiac differentiation, hiPSCs were plated on hESC-qualified Matrigel (BD Biosciences) and grown until 80–90% confluence was reached. Differentiation into cardiomyocytes (hiPSC-CMs) was carried out using a GSK3B/Wnt-signaling modulation protocol with small molecules as previously described[52], with minor modifications. Following differentiation, hiPSC-CMs were selected and purified by glucose starvation, yielding a final purity >90% hiPSC-CMs[53]. After selection, cells were cryopreserved in liquid nitrogen between day 9 and day 16 in BamBanker (Nippon Genetics). Before the experiments, cryopreserved hiPSC-CMs were thawed and expanded according to published methods[54]. HiPSC-CM were then maintained in a customized medium (RBK1), prepared by supplementing RPMI 1640 medium (Euroclone) with 1× B27 Supplement (Thermo Fisher Scientific) and 1% KnockOut Serum Replacement (Thermo Fisher Scientific).

### Biohybrid OECT and cell planting
The biohybrid OECT chips were fabricated using a process similar to that for OECM fabrication, albeit exclusively employing the M1 and P2 procedures given their planar architecture. A biohybrid OECT chip consists of an array of 30 BBL channels ($W = L = 10$ µm, $d \approx 20$ nm), wherein the compact channel size can benefit the capacitive coupling of hiPSC-CM AP to the channel, enhancing the modulation of channel resistance. An Ag/AgCl pellet was used as the non-polarizable gate to measure the standard transfer curves of the biohybrid OECT. A sterilized Pt coil was used as the common floating gate for all the channels during in vitro experiments.

Before cell planting, the OECT chips were sterilized by washing with water, 70% ethanol, and 100% ethanol (repeated twice), and then dried in a laminar-flow hood for 5 min. The chip surface was then covered with a fibronectin bovine protein solution (Merck, 40 μg/mL in PBS) for 1 h under a humidified atmosphere at 37 °C to form a fibronectin coating. After removing the solution, 10 μL RBK1 medium (supplemented with Revitacell, Thermo Fisher Scientific) carrying hiPSC-CMs was drop-cast and plated at a density of $5 \times 10^5$ cells cm$^{-2}$. The chips, together with planted cells, were then incubated for 1 h under a humidified atmosphere at 37 °C, with 5% $CO_2$, to facilitate cell spreading and the formation of a monolayer of hiPSC-CMs. After that, the RBK1 medium was gently supplemented and continuously refreshed until proceeding to tests.

### AP synchronization from hiPSC-CM to OECM
The J-inverter was formed by connecting one of the BBL channels to a load resistor ($R_J$) in series through circuit wiring. In vitro J-inverter recording was performed using a parameter analyzer (B1500A, Keysight), with two terminals supplying $V_{in}$ and $E_J$, and one terminal recording $V_{out}$. Keithley 4200A-SCS was then employed to supply the recorded $V_{out}$ into an OECM, while simultaneously biasing its Na/Ca and K channels and recording $V_{mem}$.

### Reporting summary
Further information on research design is available in the Nature Portfolio Reporting Summary linked to this article.

### Data availability
The data generated in this study have been deposited in the Zenodo database under accession code: https://doi.org/10.5281/zenodo.19389071.

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

## Acknowledgements

This work was financially supported by the Knut and Alice Wallenberg Foundation (2021.0058 and Wallenberg Initiative Materials Science for Sustainability WISE), the Swedish Research Council (2020-03243, 2022-04053, 2022-04553, 2024-04871), the European Research Council through the ERC Consolidator Grant project INFER (101125879), the European Commission through the FET-OPEN project MITICS (964677), the Pathfinder OPEN project ICONIC (101129638), and the MSCA-2023-PF project S-OECN (101152690), the Swedish Foundation for Strategic Research (IS24-0162), VINNOVA (2023-01337), and the Swedish Government Strategic Research Area in Materials Science on Functional Materials at Linköping University (Faculty Grant SFO-Mat-LiU 2009-00971). L.S. acknowledges support from NextGenerationEU - "Funding projects presented by young researchers" Project No. H45E22001210006. P.J.S. acknowledges support from the Leducq Foundation for Cardiovascular Research [18CVD05] 'Towards Precision Medicine with Human iPSCs for Cardiac Channelopathies'. A.Kh. acknowledges support from Europe (HORIZON-MSCA-2022-PF-01 PREPARE No. 101105561).

## Author contributions

D.G. and S.F. conceptualized the cardiomorphic device and designed the project. D.G. fabricated and characterized the OECM chips and performed SPICE simulation. D.T. guided circuit design and SPICE simulation. C.-Y.Y. assisted OECM chip design and fabrication. J.J. and M.X. assisted OECM fabrication. W.J. and U.B. assisted OECM characterization. H.-Y.W. and C.-Y.Y. synthesized BBL. A.W.E. synthesized Pg$_3$T-TT under I.M.'s supervision. A.Kh. and L.S. cultured and plated the hiPSC-CMs under P.J.S.'s supervision. S.D.P. and D.G. designed and fabricated the biohybrid OECT chips. A.Ky., S.D.P., and D.G. performed the in vitro recording of hiPSC-CM using the J-inverter, under M.C.'s supervision. S.D.P. performed the micrography of the biohybrid OECT. M.B. contributed to scientific discussions. S.F. supervised the project. D.G. analyzed and visualized the data. D.G. and S.F. prepared the manuscript with inputs from all authors.

## Funding

## Competing interests

The authors declare no competing interests.
