## [Transparent Peer Review file · Nature Communications]

An Organic Artificial Cardiomyocyte

Corresponding Author: Professor Simone Fabiano

Version 0:

Reviewer comments:

Reviewer #1

(Remarks to the Author)

The manuscript by Goa et al. describes the development of a circuit based on organic electrochemical transistors used to mimic ion channels of ventricular cardiomyocytes. The circuit successfully replicates electrophysiological phenomena (the action potential) of the cardiomyocytes by replicating the potassium current and the Sodium and Calcium current. It also adapts beating signature according to changes in the electrolyte concentrations, which is used to mirror particular imbalances in the cardiac physiological environment. The particular clinical relevant cases examined are hyperkalemia, hypokalemia and cardiac hypoxia (and supplementary hyper-/hyponatremia). Additionally, part of the circuit can be replaced by a biointerfacing organic electrochemical transistor to illustrate the possibility of synchronizing the artificial cardiomyocyte with the biological one.

The work presented is exciting, timely and very topical, and nicely fits in a larger current trend towards smart biohybrid systems, that both assist in optimising current systems as well as help identify and clarify the mechanisms in biological systems. I believe the work should be published in this journal after addressing a few minor comments:

The phases in Fig.1(f) the AP graph could be annotated (possibly in caption) to indicate what the number represent explicitly.

Missing caption at Suppl. fig. 4.

Could the authors explain the characteristic differences between the AP profiles shown in Fig1(g) and Suppl. Fig12 (10uA,20ms)

- stronger repolarization
- baseline shift $\sim 0\text{ V} \rightarrow 0.1\text{ V}$

“Notably, the generation of a complete OEEM AP necessitates an adequate lin stimulus. Insufficient charge delivery results in a subthreshold spike, whereas excessive charging prevents repolarization”

The chosen standard operating conditions (10uA for 20ms) seem to be chosen to fit the electrophysiological vCM AP. This is perfectly fine, and is also stated as such. However, since the effective biopotential input is given as a voltage reading (V_{out}) it might be interesting to also show the transient analysis of the OEEM AP as a function of voltage step inputs with controlled V_s input while monitoring the V_{mem} output. This could allow for a better side by side comparison of the biotic and standalone OEEM. Alternatively, the current injection from source EJ could be measured to illustrate the comparison. This would also allow for a more detailed analysis of the output V_{mem} of the biotic OEEM which seems to operate with a potential offset compared to the stand-alone OEEM.

Finally, since the conclusion mentions possible in-vivo directions, it could be useful to briefly indicate the challenges that such a system might face.

Reviewer #2

(Remarks to the Author)

In this manuscript, Gao et al. report an organic artificial cardiomyocyte based on a well-designed circuit comprising multiple organic electrochemical transistors (OECTs) with distinct operating speeds. This design closely emulates the Beeler-Reuter

model and fundamentally captures the key features of human ventricular action potentials, enabling biorealistic responses, electrical/chemical modulation capabilities, and synchronization with interfaced living cells. This work makes valuable contributions to biorealistic hardware and bioelectronics, as the device not only passively interfaces with biological tissues but also actively participates in biological processes via a similar interactive mechanism. Therefore, the reviewer recommends this paper for publication in Nature Communications following revisions addressing the below concerns:

1. A key concern affecting the manuscript's accessibility is its overall lack of clarity for broad readership. Not all (or only a small subset of) potential readers are familiar with individual OECTs or OECT-based inverters, let alone the dynamic, interdependent combination of these components in the proposed circuit. The authors should revise the explanation of the circuit's working principle to be clearer and navigable, especially for Figures 1 and 2. Possible improvements include: (i) delineating the key function of each building block and the overall circuit workflow in the main text, while leaving detailed analyses to SI; (ii) optimizing figures for clarity. For example, Figure S5a in the SI is more intuitive than Figure 1e in the main text and could be adapted or integrated to enhance readability.
2. The artificial Na and K channels incorporate their respective ions in the operating electrolytes, but the artificial Ca channel does not involve Ca^{2+} at all. Instead, the authors rely on the hysteresis of the Na channel to mimic the Ca current. This design compromises biorealism. Could Ca ions be incorporated into the Ca-channel design to better recapitulate biological reality?
3. Numerous abbreviations in the figures remain unexplained, which may confuse readers. For instance, $I_{\text{Ca(L)}}$, J_{ref} , I_{to} , I_{Kr} , I_{Ks} , and I_{K1} in Figure 1f are not defined. All abbreviations should be clarified either in the figure legends or in the main text.
4. Compared with typical artificial neuron circuits (e.g. 10.1038/s41467-022-28483-6), the inverter operation in this manuscript shows some differences: while artificial neuron circuits use two cascaded inverters with a common, fixed voltage supply (high = VDD, low = 0), the present work employs two inverters with distinct, dynamic supplies (S-inverter: high = E_{s} , low = 0; Na/Ca channel: high = $ENC \neq E_{\text{s}}$, low = V_{mem} , which is a non-constant value). What were the core design considerations behind these modifications? How do these changes reflect the electrophysiological differences between cardiomyocytes and neurons? A comparative discussion with prior work would facilitate readers' understanding of the new mechanisms.
5. The authors state that two strategies are used to achieve a slow K channel: low ionic concentration and additional RC elements. However, the device structure of K-channel OECT is also different from the Na/Ca-channel inverter; the former is a planar device, while the latter adopts a stair-type device structure. How does this structural difference influence the response kinetics, and to what extent does it synergize with the low ionic concentration and RC elements to achieve the slow K-channel response?
6. In Figure 2a, what is the mechanistic origin of the I_{Na} (or I_{Ca}) spike? What is the difference between the "steady-state" and "transient" responses?
7. The authors illustrate current changes with varying V_{mem} in Figure 2, yet in Figure 1g, V_{mem} is presented as the circuit's output that mimics vCM action potentials. Thus, it remains unclear how Figure 2 correlates with the circuit's operating process and output. Overall, the reviewer does not fully grasp the working principle of the circuits shown in Figure 2 and S5. Additional clarification on the relationship between these figures and the overall circuit function is needed.
8. The authors claim the plateau phase is established by "a secondary equilibrium of OECM," but this description is vague and requires further explanation. Furthermore, what factors disrupt this equilibrium to initiate the plateau decline?
9. The voltage profile generated by the hiPSC-CMs in Figure 5e appears distinct from the "standard" ventricular AP profile illustrated in Figure 1f. What accounts for this discrepancy? Is it related to hiPSC-CM maturity, experimental conditions, or signal processing via the J-inverter?
10. For the cell-integration experiments, how about the biocompatibility of the device? How about the device long-term stability in the cell culture environment?

Reviewer #3

(Remarks to the Author)

This paper presents an Organic Electrochemical Cardiomyocyte (OECM), a form of "cardiac-morphology wetware." It leverages the ion-electron coupling characteristics of OECTs to mimic the key electrophysiological functions of ventricular cardiomyocytes (vCMs) at the hardware level. While there are many OECTs mimicking neurons (also some from the same group), mimicking CMs are rare and worth exploring for bionic bioelectronics. The integration of OECT physics with cardiac electrophysiology is promising, and the manuscript is well-organized. I recommend that the manuscript be considered for publication in Nat Commun after addressing the following concerns.

1. What is the major challenge in mimicking CMs than Neuron? Or the authors can simply transfer the same know-hows to a new system.
2. On page 4, line 83, the paper states that the threshold voltage (V_{th}) for both BBL-based and P(g32T-TT)-based OECTs is comparable. However, the reference only cites the parameters for BBL in reference 20. Please provide a complete reference

for the P(g32T-TT) material. Additionally, since the V_{th} parameter can be influenced by the geometry of the channel, how does the authors adjust V_{th} ?

3. In the section on OECM architecture and its analogy to vCMs: The sensing inverter (S-inverter) preceding the Na/Ca channel is implemented using a planar OECT structure (Fig. 2e) that exhibits a relatively slow transient response (Supplementary Fig. 2). One question that arises is whether the switching speed of this S-inverter is sufficient to fully leverage the rapid performance of the vertical OECTs in the Na/Ca channel, or if the overall system performance is ultimately constrained by this driver stage.

4. In the section on ionic and pH modulation, the paper states that the transconductance of the BBL-based OECT is positively correlated with the ionic strength of its gating electrolyte. Is this effect specific to K^+ ions? If not, would incorporating an ion-selective membrane in the channel be a more suitable strategy?

5. In the section on ionic and pH modulation, the study demonstrates pH modulation within a relatively narrow range (pH 6–7) as shown in Supplementary Fig. 17. Expanding this detection capability would be highly beneficial.

6. Given its intended application as an artificial cardiomyocyte, the OECM is expected to undergo continuous and repetitive cycling. The manuscript would be significantly strengthened by the inclusion of data on the device's long-term operational stability and cycling endurance.

7. Any plan to interact with a living CMs? Can it work? What are the limitations.

Reviewer #4

(Remarks to the Author)

This manuscript reports a carefully executed study introducing an organic electrochemical circuit which emulates a cardiomyocyte, capable of reproducing key electrophysiological features of ventricular action potentials. The authors convincingly map ion-dependent OECT dynamics onto biophysically meaningful Na/Ca and K channel analogues, achieving realistic action-potential phases, refractoriness, and tunability. The experimental validation is convincing, and the mechanistic interpretation is sound. Overall, this work represents a significant advance in organic neuromorphic electronics, and I recommend acceptance without further revision.

Version 1:

Reviewer comments:

Reviewer #1

(Remarks to the Author)

The authors have successfully addressed all my comments. I believe the manuscript can now be accepted.

Reviewer #2

(Remarks to the Author)

The authors have adequately addressed my comments, and I have no further concerns. I recommend acceptance of the revised manuscript.

Reviewer #3

(Remarks to the Author)

The authors have addressed most of my concerns. The manuscript is in good shape towards publication.

Reviewer #5

(Remarks to the Author)

Response to the Reviewers

Dear referees, we found your reviews to be very thoughtful, and the comments were extremely helpful in enhancing the quality and thus the impact of our manuscript. Below, please find our point-by-point response in red lettering to your concerns and a description of how and where revisions to the manuscript have been made.

Reviewer #1 (Remarks to the Author):

The manuscript by Gao et al. describes the development of a circuit based on organic electrochemical transistors used to mimic ion channels of ventricular cardiomyocytes. The circuit successfully replicates electrophysiological phenomena (the action potential) of the cardiomyocytes by replicating the potassium current and the Sodium and Calcium current. It also adapts beating signature according to changes in the electrolyte concentrations, which is used to mirror particular imbalances in the cardiac physiological environment. The particular clinical relevant cases examined are hyperkalemia, hypokalemia and cardiac hypoxia (and supplementary hyper-/hyponatremia). Additionally, part of the circuit can be replaced by a biointerfacing organic electrochemical transistor to illustrate the possibility of synchronizing the artificial cardiomyocyte with the biological one. The work presented is exciting, timely and very topical, and nicely fits in a larger current trend towards smart biohybrid systems, that both assist in optimising current systems as well as help identify and clarify the mechanisms in biological systems. I believe the work should be published in this journal after addressing a few minor comments:

We thank the reviewer for their positive commentary on our manuscript and for acknowledging both its impact and novelty. In the following, we address his/her remarks:

1) *The phases in Fig. 1(f) the AP graph could be annotated (possibly in caption) to indicate what the numbers represent explicitly.*

The captions of **Fig. 1f** and **1g** have been updated to define the numbers explicitly.

2) *Missing caption at Suppl. fig. 4.*

The caption of **Supplementary Fig. 4** has been updated with more details.

3) *Could the authors explain the characteristic differences between the AP profiles shown in Fig 1(g) and Suppl. Fig 12 (10 uA, 20ms) - stronger repolarization - baseline shift ~ 0 V \rightarrow 0.1 V*

Excellent comment. The differences in the OEEM AP profiles observed across figures and tests arise from variations in channel drive strength between fabricated OEEM chips. The current fabrication process, which involves Parylene C patterning and OMIEC spinning coating, inevitably introduces small variations in the channel dimensions and thickness. For instance, a K channel with slightly lower BBL thickness exhibits higher resistance when turned off (phase 4) and a weaker discharging strength when turned on. To accommodate these device-to-device variations, we adjust E_K within a range of -0.05 to -0.15 V while keeping all other parameters constant to ensure that each OEEM can function properly and generate APs with a standard APD of approximately 300 ms. This strategy yields reproducible APD but shifts the baseline. In the above example, the lower conductance in K channel requires more negative E_K to drive the OEEM, leading to a lower resting potential in V_{mem} and stronger repolarization in phase 4. The above analysis has

been provided in the caption of **Fig. 1** and **Method – Electrical characterization**, and we further labeled the E_K value used for each specific test in the corresponding figure captions.

4) “Notably, the generation of a complete OEEM AP necessitates an adequate I_{in} stimulus. Insufficient charge delivery results in a subthreshold spike, whereas excessive charging prevents repolarization”. The chosen standard operating conditions (10uA for 20ms) seem to be chosen to fit the electrophysiological vCM AP. This is perfectly fine, and is also stated as such. However, since the effective biopotential input is given as a voltage reading (V_{out}) it might be interesting to also show the transient analysis of the OEEM AP as a function of voltage step inputs with controlled V_S input while monitoring the V_{mem} output. This could allow for a better side by side comparison of the biotic and standalone OEEM. Alternatively, the current injection from source EJ could be measured to illustrate the comparison. This would also allow for a more detailed analysis of the output V_{mem} of the biotic OEEM which seems to operate with a potential offset compared to the stand-alone OEEM.

We appreciate the reviewer’s clear understanding of the different operation modes between standalone and biotic OEEMs. In a standalone OEEM, the S-inverter is constructed from a pair of complementary OEETs and features a high voltage gain at the transition voltage (**Fig. 2c**). Upon I_{in} stimulation, the S-inverter rapidly sweeps V_S from 0.6 V to ~0.3 V during Phase 0 (see **Supplementary Fig. 4**), thereby effectively modulating the Na/Ca channel across its resting, activation, and inactivation states. In contrast, the J-inverter in a biotic OEEM comprises a load resistor in series with an n-type BBL OEET. This configuration has a much lower voltage gain than the S-inverter and is governed by the APs generated by hiPSC-CMs rather than by electrical instruments (**Supplementary Fig. 23**). The hiPSC-CMs can generate ventricular APs with full amplitude of ~120 mV, yet only ~30 mV is effectively coupled to the BBL channel due to suboptimal coupling between the hiPSC-CMs and the BBL channel in the biohybrid OEET. As a result, the firing of hiPSC-CMs produces a small V_{out} swing, from 0.5 V to 0.45 V (**Fig. 5f**), leading to less effective modulation of the Na/Ca channel opening and closing. Overall, the differences in voltage gain and output swing between the S- and J-inverters lead to different AP profiles (resting potential, shape, etc.). To bring the behavior of the biotic OEEM closer to that of the standalone one, several strategies could be considered **1**) increasing the transconductance of the biohybrid BBL OEET (e.g. by using vertical OEET architectures); **2**) improving the bioelectrical coupling between hiPSC-CMs and the BBL channel; and **3**) incorporating a non-inverting amplification stage (e.g., two cascaded OEET-based inverters) between the J-inverter and the Na/Ca channel to increase the voltage swing.

As suggested by the reviewer, we also performed control experiments using a standalone OEEM with externally controlled V_S inputs. We applied V_S steps of different amplitudes and recorded V_{mem} , I_{NC} , and I_K (**Supplementary Fig. 25**). A larger step (0.6-0.3 V) yielded APs similar to those produced by standalone OEEMs, while a smaller step (0.5-0.4 V) yielded APs akin to those of the biotic OEEMs. These control experiments further confirm that the different amplitudes in the output swing of the S- and J-inverters underpin the AP shape differences between standalone and biotic OEEMs.

5) Finally, since the conclusion mentions possible in-vivo directions, it could be useful to briefly indicate the challenges that such a system might face.

In the updated **Conclusions**, we briefly discussed the challenges that should be addressed when translating rigid OEEM chips for *in vivo* implants, including biocompatibility, stability, mechanical matching, and power management strategies.

Reviewer #2 (Remarks to the Author):

In this manuscript, Gao et al. report an organic artificial cardiomyocyte based on a well-designed circuit comprising multiple organic electrochemical transistors (OECTs) with distinct operating speeds. This design closely emulates the Beeler-Reuter model and fundamentally captures the key features of human ventricular action potentials, enabling biorealistic responses, electrical/chemical modulation capabilities, and synchronization with interfaced living cells. This work makes valuable contributions to biorealistic hardware and bioelectronics, as the device not only passively interfaces with biological tissues but also actively participates in biological processes via a similar interactive mechanism. Therefore, the reviewer recommends this paper for publication in Nature Communications following revisions addressing the below concerns:

We thank the reviewer for their positive commentary on our manuscript and for acknowledging both its impact and novelty. In the following, we address his/her remarks:

1) A key concern affecting the manuscript's accessibility is its overall lack of clarity for broad readership. Not all (or only a small subset of) potential readers are familiar with individual OECTs or OECT-based inverters, let alone the dynamic, interdependent combination of these components in the proposed circuit. The authors should revise the explanation of the circuit's working principle to be clearer and navigable, especially for Figures 1 and 2. Possible improvements include: (i) delineating the key function of each building block and the overall circuit workflow in the main text, while leaving detailed analyses to SI; (ii) optimizing figures for clarity. For example, Figure S5a in the SI is more intuitive than Figure 1e in the main text and could be adapted or integrated to enhance readability.

We appreciate the reviewer's valuable suggestions on how to improve the clarity of our work for a broad readership. In the main text, **OECM architecture** section, we incorporated a distinct paragraph to explain the key function of each building block, and discussed how these blocks are associated in the OECM circuit. We also revised **Fig. 2** by adding two sets of schematic circuits that depict the testing conditions for both steady-state OECM characterization and real-time OECM operation (transient response). We linked the circuits (**Fig. 2a-b**) to the data plots (**Fig. 2c-e**) by addressing the electrical parameters with colored labels so that the readers can better understand how these data were recorded. Furthermore, we updated the caption of **Fig. 2** to explain the differences between steady-state and real-time (transient) characterizations.

2) The artificial Na and K channels incorporate their respective ions in the operating electrolytes, but the artificial Ca channel does not involve Ca²⁺ at all. Instead, the authors rely on the hysteresis of the Na channel to mimic the Ca current. This design compromises biorealism. Could Ca ions be incorporated into the Ca-channel design to better recapitulate biological reality?

We appreciate the reviewer's suggestion and the opportunity to clarify this point. The OECM is designed to emulate the vCM's bioelectrical excitability rather than to reproduce its biological ion-channel structures. In our architecture, there is no separate artificial Ca channel. Instead, the OECM employs a single mixed charging channel (the Na/Ca inverter) that supplies both the fast sodium-like inward current (I_{Na}) and the slow calcium-like inward current (I_{Ca}). These two current components arise from the forward and backward voltage swings of the inverter, which reproduce the temporal roles of Na and Ca currents in shaping the AP. Both currents are generated when the OECTs in the Na/Ca channel are turned on, which requires the channel materials to be electrochemically doped by ions from the electrolyte. Thus, the device operation depends on the ionic conductivity of the electrolyte rather than on the presence of specific ion species. In the current OECM architecture, we use a 140 mM NaCl aqueous electrolyte to provide sufficient ionic strength for stable electrochemical doping.

Nevertheless, following the reviewer's suggestion, we examined whether incorporating Ca²⁺ ions would affect device behavior. We prepared an electrolyte solution containing 140 mM NaCl and 2 mM CaCl₂,

corresponding to the physiological extracellular calcium concentration. The Na/Ca channel exhibited comparable current-transfer characteristics in both the mixed NaCl/CaCl₂ (140/2 mM) electrolyte and the original NaCl (140 mM) electrolyte, and the resulting OECMs generated nearly identical APs.

Fig. R1. Comparison of AP shape with a) 140 mM NaCl and b) 140 mM NaCl and 2 mM CaCl₂ in the Na/Ca channel OECTs gating electrolyte.

3) Numerous abbreviations in the figures remain unexplained, which may confuse readers. For instance, *ICa(L)*, *Jref*, *Ito*, *IKr*, *IKs*, and *IK1* in Figure 1f are not defined. All abbreviations should be clarified either in the figure legends or in the main text.

Excellent comment. The ion channel currents in Fig. 1f are explained with full names in the caption.

4) Compared with typical artificial neuron circuits (e.g. 10.1038/s41467-022-28483-6), the inverter operation in this manuscript shows some differences: while artificial neuron circuits use two cascaded inverters with a common, fixed voltage supply (high = V_{DD} , low = 0), the present work employs two inverters with distinct, dynamic supplies (*S*-inverter: high = E_s , low = 0; Na/Ca channel: high = $ENC \neq E_s$, low = V_{mem} , which is a non-constant value). What were the core design considerations behind these modifications? How do these changes reflect the electrophysiological differences between cardiomyocytes and neurons? A comparative discussion with prior work would facilitate readers' understanding of the new mechanisms.

We have prepared **Supplementary Note 3** to comprehensively compare the similarities and differences between the OECM and OECN architectures. We explained why different supply rails are adopted in these architectures, and analyzed the core design principles that differentiate the OECM from existing OECNs.

5) The authors state that two strategies are used to achieve a slow *K* channel: low ionic concentration and additional RC elements. However, the device structure of *K*-channel OECT is also different from the Na/Ca-channel inverter; the former is a planar device, while the latter adopts a stair-type device structure. How does this structural difference influence the response kinetics, and to what extent does it synergize with the low ionic concentration and RC elements to achieve the slow *K*-channel response?

Again, excellent comment. The stair-type architecture used in the Na/Ca inverter was introduced primarily to reduce the OECTs' channel length to $\sim 0.8 \mu\text{m}$. This short channel length enables high transconductance and allows the Na/Ca inverter to deliver sufficient charging capacity for OECM operation. Achieving sub-micron channel length with planar OECT geometries has proven challenging with our current fabrication process. The stair-type design allows us to translate the thickness of the insulating Parylene C layer into the

effective channel length. In principle, these stair-type devices could be replaced with planar OECTs if sub-micron channel lengths could be reliably achieved using planar fabrication.

Importantly, the device topology itself (planar vs stair-type) does not directly influence the response kinetics. Instead, the switching speed of an OECT is primarily governed by the channel dimensions (length, width, and thickness), which determine the channel's volumetric capacitance and thus the characteristic device's charging time. The slow response of the K channel arises from the combined effect of external ($R_{dK}C_{dK}$) and internal ($R_iC_{ch,K}$) RC elements, as described in **Supplementary Note 2**:

“The transient response of the K channel is governed by two distinct RC time constants existing at different levels of the circuitry. The first time constant arises from the electronic RC components that are directly coupled to the gate of the K-OECT, resulting in a delay characterized by $R_{dK}C_{dK} \approx 47$ ms. The second time constant, denoted as $\tau_{on,K} = R_iC_{ch,K}$, originates from the ionic circuit within the K-OECT. Here, R_i represents the ionic resistance of the electrolyte extending from the gate to the channel, while $C_{ch,K}$ refers to the volumetric capacitance of the K-OECT's channel.”

For the K-channel OECT in the OECM, the channel dimensions are $L = 6 \mu\text{m}$, $W = 600 \mu\text{m}$, $d \approx 20 \text{ nm}$. Using the reported volumetric capacitance of BBL ($C^* = 725 \text{ F cm}^{-3}$, 10.1038/s44460-025-00007-x), the channel capacitance can be estimated to be $C_{ch,K} = LWdC^* \approx 52 \text{ nF}$. To further evaluate the contribution of R_i and $C_{ch,K}$ on the K-channel OECT's intrinsic response time (without external RC), we measured the time constant (τ_{on}) with different electrolyte concentrations and channel dimensions. These results, now included in **Supplementary Fig. 7**, confirm that a higher R_i (lower electrolyte concentration) or larger $C_{ch,K}$ (larger channel dimension) leads to longer response time.

6) In Figure 2a, what is the mechanistic origin of the I_{Na} (or I_{Ca}) spike? What is the difference between the “steady-state” and “transient” responses?

We thank the reviewer for the opportunity to clarify this point. The top panel in **Fig. 2c** (previously Fig. 2a) shows the charge-transfer characteristics of the Na/Ca channel during charging. In this measurement, V_{mem} is swept from 0 V to 0.6 V and then back to 0 V under instrumental control, while the current through the Na/Ca channel is recorded. The currents measured during the forward and backward sweeps are denoted as I_{Na} and I_{Ca} , respectively. These traces therefore represent the evolution of the Na/Ca channel current in response to a controlled V_{mem} sweep. When V_{mem} is far from 0.3 V, the Na/Ca channel remains off and the current is low. As V_{mem} approaches ~ 0.3 V, the S-inverter rapidly switches state and turns on the Na/Ca channel, producing a sharp increase in current that appears as a spike in the I_{Na} and I_{Ca} traces.

Regarding the terminology used in the manuscript, “steady-states” characterization refers to the standard electrical measurements commonly used to benchmark the properties of OECTs and OECT-based inverters (10.1039/D2CS00920J). It probes the devices' performance at equilibrium and is independent of time. Such “steady-state” tests are performed prior to operating an OECM to verify that the individual circuit elements function correctly. By contrast, the operation of the OECM itself is inherently time-dependent because the system generates oscillating APs in real time. Therefore, the relevant quantities (currents and voltages) must be recorded as functions of time (**Fig. 1g**), which we refer to as “transient” responses. In **Fig. 2c**, the steady-state behaviors of I_{Na} , I_{Ca} , I_K , and V_S are plotted as functions of V_{mem} . In **Fig. 2d-e**, the transient responses of I_{in} , I_{Na} , I_{Ca} , I_K , V_{mem} , and V_S are recorded as functions of time during OECM operation. Using these time-resolved data, we reconstructed the relationships between these electrical parameters by plotting them against V_{mem} . In this sense, **Fig. 2d-e** can be viewed as an alternative representation of the data shown in **Fig. 1g**. Specifically, **Fig. 2d** presents the fast forward V_{mem} swing, while **Fig. 2e** presents the slower backward V_{mem} swing. To clarify this distinction, we have revised the caption of **Fig. 2** to include more detailed descriptions of the testing conditions for both “steady-state” and “real-time” characterizations.

7) The authors illustrate current changes with varying V_{mem} in Figure 2, yet in Figure 1g, V_{mem} is presented as the circuit's output that mimics vCM action potentials. Thus, it remains unclear how Figure 2 correlates with the circuit's operating process and output. Overall, the reviewer does not fully grasp the working principle of the circuits shown in Figure 2 and S5. Additional clarification on the relationship between these figures and the overall circuit function is needed.

We agree with the reviewer that the data presentation in **Fig. 2** lacked clarity in our first submission. We have therefore updated **Fig. 2** together with its caption to clarify how and why these tests were carried out. With these updates, we hope the reviewer and readers can interpret these figures more easily.

8) The authors claim the plateau phase is established by “a secondary equilibrium of OEEM,” but this description is vague and requires further explanation. Furthermore, what factors disrupt this equilibrium to initiate the plateau decline?

Excellent comment. In excitable biological cells, “equilibrium” typically refers to the resting potential (E_{rest}), where the sum of all ionic currents is zero. In vCM, however, the AP includes a quasi-stationary plateau phase (Phase 2) in which inward and outward currents transiently balance at a depolarized potential. We use the term secondary equilibrium to describe this temporary balance. In the OEEM, this plateau arises from the balance between the inward and outward currents supplied by the Na/Ca channel and the outward current of the K channel. During phase 2, the S-inverter's output (V_S) biases the Na/Ca channel into a state of high conductance, producing a sustained inward “charging” current. In parallel, the K channel provides an outward “discharging” current. When these two currents are equal, the membrane potential stabilizes at a depolarized level, giving rise to the plateau. The balance is not permanent because the K channel continues to evolve over time. The K-OECT exhibits slow ionic doping kinetics with a relatively long time constant, and its conductance gradually increases as ions accumulate in the BBL channel. As a result, the outward current progressively grows during the plateau phase. Once the K current exceeds the inward Na/Ca current, the membrane potential begins to decrease. This decrease in V_{mem} feeds back through the S-inverter, shifting the Na/Ca channel away from its conductance peak and rapidly reducing the inward current. The resulting imbalance terminates the plateau (Phase 2) and initiates repolarization (Phase 3).

To clarify this mechanism, we have added a detailed explanation as **Supplementary Note 4**.

9) The voltage profile generated by the hiPSC-CMs in Figure 5e appears distinct from the “standard” ventricular AP profile illustrated in Figure 1f. What accounts for this discrepancy? Is it related to hiPSC-CM maturity, experimental conditions, or signal processing via the J-inverter?

We thank the reviewer for this important question. The discrepancy primarily originates from the limited voltage gain of the J-inverter rather than from the intrinsic electrophysiology of the hiPSC-CMs. In a standalone OEEM, the Na/Ca channel is controlled by an S-inverter composed of a pair of complementary OECTs, which exhibits high voltage gain near its transition voltage (**Fig. 2c**). Upon I_{in} stimulation, the S-inverter rapidly sweeps V_S from 0.6 V to \sim 0.3 V during Phase 0 (**Supplementary Fig. 4**), thus effectively modulating the Na/Ca channel across its resting, activation, and inactivation states. In contrast, the J-inverter used in the biotic OEEM consists of a load resistor in series with a BBL OECT and therefore exhibits much lower voltage gain. In this configuration, the inverter is driven by the APs from hiPSC-CMs rather than by an electrical stimulus (see **Supplementary Fig. 23**). Although the hiPSC-CMs can generate ventricular APs with amplitudes of \sim 120 mV, only \sim 30 mV is effectively coupled to the BBL channel due to suboptimal coupling between the hiPSC-CMs and the BBL channel in the biohybrid OECT. As a result, the output voltage swing is limited (from 0.5 V to 0.45 V; **Fig. 5f**), leading to weaker modulation of the Na/Ca channel and consequently a different AP profile.

To verify this interpretation, we performed control experiments using a standalone OEEM with externally applied V_S steps of different amplitudes and recorded V_{mem} , I_{NC} , and I_K (**Supplementary Fig. 25**). A larger

step (0.6-0.3 V) yielded APs similar to those observed in standalone OECMs, while a smaller step (0.5-0.4 V) generated APs resembling those obtained with biotic OECMs. These results confirm that the reduced output swing of the J-inverter is responsible for the observed difference in AP shape between the standalone and biotic OECMs.

Potential strategies to bring the biotic OECM behavior closer to that of the standalone OECM include **1)** increasing the transconductance of the biohybrid BBL OECT (e.g., using vertical OECT architectures); **2)** improving the bioelectrical coupling between hiPSC-CMs and the BBL channel; or **3)** incorporating a non-inverting element (e.g., two cascaded OECT-based inverters) between the J-inverter and the Na/Ca channel to amplify the voltage swing.

10) For the cell-integration experiments, how about the biocompatibility of the device? How about the device long-term stability in the cell culture environment?

In the cell-integration experiments, hiPSC-CMs interface only with the sensing BBL-based OECT, while the remaining components of the OECM circuit are not in direct contact with the cells. Therefore, the biocompatibility and stability of the system are primarily determined by the properties of the BBL films. Recently, we comprehensively evaluated the chemical and morphological stability of BBL in cell culture media (10.1002/sml.202404451). The results show that BBL films are highly stable under biological conditions and exhibit negligible degradation during cell culture experiments. In the same study, we also assessed the biocompatibility of BBL, showing high cell viability (> 93% for HT22 cells) and normal cell proliferation comparable to control substrates, confirming the non-cytotoxic nature of the material. In addition, the nanoscale surface roughness of BBL films (9.3 ± 1.0 nm) was found to promote focal adhesion formation and improve cell spreading. These results indicate that BBL provides a stable and biocompatible interface for cell-device integration. We have added a discussion of this point in the revised manuscript and cited the above study on page 10, line 264.

Reviewer #3 (Remarks to the Author):

This paper presents an Organic Electrochemical Cardiomyocyte (OECM), a form of "cardiac-morphology wetware." It leverages the ion-electron coupling characteristics of OECTs to mimic the key electrophysiological functions of ventricular cardiomyocytes (vCMs) at the hardware level. While there are many OECTs mimicking neurons (also some from the same group), mimicking CMs are rare and worth exploring for bionic bioelectronics. The integration of OECT physics with cardiac electrophysiology is promising, and the manuscript is well-organized. I recommend that the manuscript be considered for publication in Nat Commun after addressing the following concerns.

We thank the reviewer for their positive commentary on our manuscript and for acknowledging both its novelty and impact. In the following, we address his/her remarks:

1) What is the major challenge in mimicking CMs than Neuron? Or the authors can simply transfer the same know-hows to a new system.

We thank the reviewer for this important question. The major challenge in mimicking vCMs, compared to neurons, lies in reproducing the characteristic plateau phase (Phase 2) of the ventricular AP. This phase requires a sustained slow inward calcium current (I_{Ca}) that balances the outward I_K , creating a quasi-stationary depolarized state. Such a mechanism is absent in neuronal APs, which typically involve only fast sodium activation followed by potassium-driven repolarization.

Our previous work on conductance-based organic electrochemical neurons (HH OECN, 10.1038/s41563-022-01450-8) provided the starting framework for this study. The HH OECN architecture consists of a fast charging Na channel and a slow discharging K channel, both connected to V_{mem} and responsible for charging or discharging C_{mem} . The Na channel in the HH OECN is implemented using a single BBL-based OECT and generates a transient sodium-like current during depolarization. In contrast, the OECM requires an additional mechanism to reproduce the calcium-mediated plateau. To achieve this, the Na/Ca channel in the OECM is implemented using an OECT-based inverter that can generate both a transient I_{Na} and a sustained I_{Ca} , which counterbalances I_{K} at the OECM's secondary equilibrium potential (~ 0.3 V). The K channels in HH OECN and OECM share a similar configuration but operate under different electrolyte conditions. While the K-OECT in the OECN uses a standard 100 mM NaCl electrolyte, the OECM employs a low ionic concentration (5 mM KCl) to increase the switching threshold and significantly slow the response. This keeps the K channel closed during the forward V_{mem} swing and allows it to open gradually during Phase 2, thereby balancing I_{Ca} and producing the plateau.

Therefore, although the HH OECN provided an initial framework, reproducing the vCM-like AP required substantial modification to incorporate calcium-like dynamics and establish the secondary equilibrium responsible for the plateau. In revisions, we have added a detailed comparison between OECN and OECM architectures in **Supplementary Note 3**, and further discussion of the primary and secondary equilibria in **Supplementary Note 4**.

2) On page 4, line 83, the paper states that the threshold voltage (V_{th}) for both BBL-based and P(g₃2T-TT)-based OECTs is comparable. However, the reference only cites the parameters for BBL in reference 20. Please provide a complete reference for the P(g₃2T-TT) material. Additionally, since the V_{th} parameter can be influenced by the geometry of the channel, how does the authors adjust V_{th} ?

On page 4, line 100, we have updated the reference to one of our latest publications (10.1038/s44460-025-00007-x), where the V_{th} of both BBL and P(g₃2T-TT) OECTs are systematically characterized. The relevant parameters are reported in Supplementary Tables 1-2 of that work. Specifically, the measured V_{th} values are approximately 0.15 V for BBL and -0.1 V P(g₃2T-TT). Although not identical (in absolute value), these values are comparable within the operating window of the OECTs and enable the construction of OECT-based complementary inverters whose transition voltage lies close to the midpoint of the supply voltage.

Regarding the influence of device geometry, in silicon MOSFETs, the V_{th} often shifts with channel scaling due to short-channel effects. In contrast, OECT operation relies on volumetric electrochemical doping of the channel material. As a result, for devices with μm -scale channel lengths, V_{th} is largely determined by the intrinsic redox properties of the channel material and is only weakly dependent on channel geometry (10.1038/s44460-025-00007-x). Therefore, OECTs fabricated with different channel geometries typically exhibit similar V_{th} values. Because the intrinsic V_{th} values of BBL and P(g₃2T-TT) are already reasonably matched, no additional adjustment of V_{th} was required for the construction of the complementary inverters used in this work.

3) In the section on OECM architecture and its analogy to vCMs: The sensing inverter (S-inverter) preceding the Na/Ca channel is implemented using a planar OECT structure (Fig. 2e) that exhibits a relatively slow transient response (Supplementary Fig. 2). One question that arises is whether the switching speed of this S-inverter is sufficient to fully leverage the rapid performance of the vertical OECTs in the Na/Ca channel, or if the overall system performance is ultimately constrained by this driver stage.

Excellent comment. In the original submission, we did not include the transient response of the S-inverter in the Supplementary Information. The previous Supplementary Fig. 2b instead showed the slow response of the discharging K channel. To address the reviewer's question, we have now characterized the transient response of the S-inverter using measurements similar to those performed for the Na/Ca channel. Although the planar OECTs in the S-inverter have a longer channel length than the vertical OECTs used in the Na/Ca

channel, they also have a smaller channel width (e.g., BBL OECT in the S-inverter: $L = 6 \mu\text{m}$, $W = 200 \mu\text{m}$; in the Na/Ca channel: $L = 0.8 \mu\text{m}$, $W = 800 \mu\text{m}$). As a result, the S-inverter exhibits a relatively small switching time constant (τ_{on} of 0.21 ms). We further evaluated the transient response of the cascaded S-inverter and Na/Ca channel and observed a similarly fast response (τ_{on} of 0.38 ms; **Supplementary Fig. 3**). These results indicate that the S-inverter switches sufficiently fast and does not limit the dynamic response of the Na/Ca channel.

4) *In the section on ionic and pH modulation, the paper states that the transconductance of the BBL-based OECT is positively correlated with the ionic strength of its gating electrolyte. Is this effect specific to K^+ ions? If not, would incorporating an ion-selective membrane in the channel be a more suitable strategy?*

We thank the reviewer for this insightful question. The operation of a BBL OECT involves two coupled half-reactions: oxidation at the Ag/AgCl gate ($\text{Ag}_{(s)} - e^- + \text{Cl}^-_{(aq)} \rightleftharpoons \text{AgCl}_{(s)}$) and reduction of the BBL channel ($\text{BBL}^0 + e^- + \text{M}^+_{(aq)} \rightleftharpoons [\text{BBL}^-\text{M}^+]$). The Nernst potentials of both processes depend on the ion concentrations in the bulk electrolyte and can be expressed as:

$$E_G = E^0 - \frac{RT}{F} \ln[\text{Cl}^-]$$

$$E_{ch} = E^0 + \frac{RT}{F} \ln\left(\frac{1-x}{x} [\text{M}^+]\right)$$

Where R , T , and F are the gas constant, temperature, and Faraday constant, respectively, and x is the doping fraction of BBL. Increasing the ion concentration (activity) of the electrolyte reduces E_G and increases E_{ch} , as the higher availability of Cl^- and M^+ makes the reactions thermodynamically more favorable. Because the OECT threshold voltage (V_{th}) is proportional to $E_G - E_{ch}$, increasing ionic strength reduces V_{th} and thus increases the transconductance ($g_m \propto V_G - V_{\text{th}}$).

Therefore, this effect is not specific to K^+ , but reflects a general dependence on the overall ionic strength of the electrolyte. In the current OECM architecture, each OECT operates with its own gating electrolyte, and the K-channel OECT is simply gated with 5 mM KCl without other ionic species. Under this condition, ion selectivity is not required for device operation. In future, more biorealistic OECM implementations where multiple OECTs operate within a shared multi-ion electrolyte, ion-selective membranes could indeed become advantageous. Such membranes could allow different channels to respond preferentially to specific ions while experiencing distinct effective ionic strengths.

5) *In the section on ionic and pH modulation, the study demonstrates pH modulation within a relatively narrow range (pH 6–7) as shown in Supplementary Fig. 17. Expanding this detection capability would be highly beneficial.*

The limited pH detection range arises from the intrinsic electrochemistry of BBL, whose conductivity relies on the injection of electrons stabilized by cations. At lower pH (<6), the high concentration of protons leads to protonation of the benzimidazole nitrogen in the BBL backbone, a process which drastically suppresses the material's conductivity (10.1038/s41563-025-02478-2). Besides, protons are significantly smaller and more mobile than K^+ . At high proton concentration, protonated BBL chains electrostatically repel the cations required to stabilize the n-doped state, thereby preventing normal K-mediated doping of the channel. As a result, the K-channel OECT can no longer provide the repolarizing current required for proper OECM operation, and the system remains in a sustained depolarized state. This mechanism, therefore, sets the lower pH limit of the device's operational window. From a physiological perspective, the pH range explored here (6-7) is highly relevant for cardiac environments. In the human heart, extracellular pH is typically ~ 7.4 but can drop to ~ 6.0 - 6.8 during acute myocardial ischemia or localized lactic acidosis. Thus, the OECM accurately mirrors this window, making it a high-fidelity tool for emulating the onset of ischemic events.

6) Given its intended application as an artificial cardiomyocyte, the OECM is expected to undergo continuous and repetitive cycling. The manuscript would be significantly strengthened by the inclusion of data on the device's long-term operational stability and cycling endurance.

Excellent point. As suggested by the reviewer, we evaluated the operational stability of the OECM by pacing the device at 60 beats per minute for 1 h, corresponding to 3,600 consecutive APs (**Supplementary Fig. 15**). Throughout this period, the OECM maintained stable and rhythmic AP generation, with the overall waveform preserved despite moderate changes in amplitude/duration (~20%). These variations primarily originate from the gradual degradation of the p-type OECTs, which slightly shifts the switching voltage of the S-inverter and reduces the charging capacity of the Na/Ca channel. The endurance of the OECM is mainly governed by the electrochemical stability of the channel materials. In our previous studies (e.g. 10.1002/adma.202106235), we showed that the n-type BBL is remarkably stable, with BBL-based OECTs sustaining pulsed gating (0-0.6 V) approaching 6 h without a noticeable reduction in I_D . In contrast, under similar conditions, the p-type P(g₃2T-TT)-based OECTs degrade rapidly (~40% I_D loss within 20 min, 10.1002/adma.202509314), which has been attributed to oxygen-induced side reactions. The long-term endurance of OECMs could be improved by suppressing these side reactions (10.1002/adfm.202302249) or by employing more electrochemically stable p-type materials, such as diketopyrrolopyrrole-based polymers (10.1038/s41586-022-05592-2).

7) Any plan to interact with a living CMs? Can it work? What are the limitations.

We thank the reviewer for this question. We note that preliminary results demonstrating interaction between an OECM and living vCMs (hiPSC-CMs) were already presented in Fig. 5. In this experiment, the hiPSC-CMs were interfaced with a biohybrid OECT so that the cells' APs could modulate the OECT conductance and eventually translate into measurable V_{out} through the J-inverter. The recorded V_{out} sequence was used to trigger the artificial APs in the OECM, demonstrating the feasibility of bioelectronic coupling of vCMs with OECMs. However, achieving fully integrated real-time coupling between living cardiomyocytes and the OECM requires reducing the output impedance of the biohybrid interface. In the current configuration, the J-inverter delivers limited current due to the relatively high resistance of the biohybrid OECT and the load resistor. One promising strategy is to increase the transconductance of the OECT (i.e., lower its on-state resistance) while maintaining a small footprint. This could be achieved by reducing the channel length, for example, to sub-micrometer dimensions, enabling the vCM to effectively seal the channel area while efficiently transducing biotic APs into conductance modulation. In such a configuration, the J-inverter would exhibit lower impedance and provide sufficient current to directly drive the OECM circuit. Future work will explore such device architectures to enable real-time vCM- OECM synchronization.

Reviewer #4 (Remarks to the Author):

This manuscript reports a carefully executed study introducing an organic electrochemical circuit which emulates a cardiomyocyte, capable of reproducing key electrophysiological features of ventricular action potentials. The authors convincingly map ion-dependent OECT dynamics onto biophysically meaningful Na/Ca and K channel analogues, achieving realistic action-potential phases, refractoriness, and tunability. The experimental validation is convincing, and the mechanistic interpretation is sound. Overall, this work represents a significant advance in organic neuromorphic electronics, and I recommend acceptance without further revision.

We thank the reviewer for their positive commentary on our manuscript and for acknowledging both its impact and novelty.